# GRAPHUNIVERSE: SYNTHETIC GRAPH GENERATION FOR EVALUATING INDUCTIVE GENERALIZATION

**Louis Van Langendonck**
Universitat Politècnica de Catalunya, Spain
louis.van.langendonck@upc.edu

**Guillermo Bernárdez**
University of California Santa Barbara, USA
guillermo_bernardez@ucsb.edu

**Nina Miolane**
University of California Santa Barbara, USA
ninamiolane@ucsb.edu

**Pere Barlet-Ros**
Universitat Politècnica de Catalunya, Spain
pere.barlet@upc.edu

## ABSTRACT

A fundamental challenge in graph learning is understanding how models generalize to new, unseen graphs. While synthetic benchmarks offer controlled settings for analysis, existing approaches are confined to single-graph, transductive settings where models train and test on the same graph structure. Addressing this gap, we introduce GraphUniverse, a framework for generating entire families of graphs to enable the first systematic evaluation of inductive generalization at scale. Our core innovation is the generation of graphs with persistent semantic communities, ensuring conceptual consistency while allowing fine-grained control over structural properties like homophily and degree distributions. This enables crucial but under-explored robustness tests, such as performance under controlled distribution shifts. Benchmarking a wide range of architectures—from GNNs to graph transformers and topological architectures—reveals that strong transductive performance is a poor predictor of inductive generalization. Furthermore, we find that robustness to distribution shift is highly sensitive not only to model architecture choice but also to the initial graph regime (e.g., high vs. low homophily). Beyond benchmarking, GraphUniverse's flexibility and scalability can facilitate the development of robust and truly generalizable architectures. The framework is open-source at https://github.com/LouisVanLangendonck/GraphUniverse.

## 1 INTRODUCTION

Graph learning has emerged as a powerful paradigm for learning from relational data across diverse domains, from drug discovery (Wong et al., 2024) and fraud detection (Cheng et al., 2025) to knowledge graphs (Galkin et al., 2023). Graph Neural Networks (GNNs) (Scarselli et al., 2008), with its countless variants (Gilmer et al., 2017; Kipf & Welling, 2017; Hamilton et al., 2017; Xu et al., 2019; Veličković et al., 2018), have demonstrated remarkable success in extending deep learning frameworks to graph-structured data, achieving competitive performance on tasks ranging from node classification to graph-level prediction. However, unlike the transformative leap from task-specific models to general-purpose architectures observed in natural language processing and computer vision, graph learning remains largely limited to specialized, task-specific models with limited evidence of robust generalization and scaling capabilities.

Recent analyses argue that progress in graph learning is hindered by a flawed benchmarking culture. Bechler-Speicher et al. (2025), for instance, critiques the field's excessive focus on incremental gains on weak benchmarks, which often fail to outperform simpler non-graph baselines. They also highlight a scarcity of large-scale, diverse datasets, arguing that these limitations hinder the development of models that can generalize and scale. Complementing this, Wang et al. (2025) pinpoints critical gaps in the theoretical understanding of model behavior—particularly concerning robustness to distribution shifts and generalization guarantees. They identify these theoretical weaknesses as key obstacles preventing graph models from advancing beyond narrow, task-specific applications.

To remedy these issues, both works propose creating better datasets through synthetic generation and quality-centric curation, alongside developing metrics for generalization, robustness, and trustworthiness. However, existing synthetic generation tools like GraphWorld (Palowitch et al., 2022) are fundamentally limited in this regard. They generate graphs as isolated, independent instances, which restricts evaluation to transductive settings where a model trains and tests on the same structure. This single-graph paradigm makes it impossible to study generalization to unseen graphs and constrains to experiment at scale—precisely the two capabilities identified as critical for building powerful graph foundation models (Wang et al., 2025).

We address this gap with **GraphUniverse**: a framework for generating graph families at scale. Our contributions can be summarized as follows:

1. We develop a hierarchical generative model extending Degree Corrected-Stochastic Block Models (DC-SBMs) (Karrer & Newman, 2011) to an inductive setting with multiple graphs that maintain semantic consistency—i.e. node identities or community structures persist across different graph instances—while enabling controlled variation in their structural properties.

2. GraphUniverse is available as a PyPI package at `https://pypi.org/project/graph-universe/`, with source code at `https://github.com/LouisVanLangendonck/GraphUniverse`. We have integrated GraphUniverse into TopoBench (Telyatnikov et al., 2025) to facilitate reproduction and extension of our experimental results.

3. We conduct systematic benchmarking comparing inductive and transductive evaluation across classical and contemporary graph architectures, revealing differences in model rankings between paradigms. Additionally, we evaluate model robustness under controlled property shifts, an analysis only possible with our inductive framework, finding that robustness strongly depends on both architecture choice and initial graph properties. These findings challenge conventional assumptions about graph model performance and demonstrate the critical importance of evaluation paradigm choice in assessing true model capabilities. Furthermore, we demonstrate that GraphUniverse-generated datasets can serve as effective proxies for real-world datasets, with model rankings showing strong correlations with those obtained on real data.

4. We provide an interactive web platform (`https://graphuniverse.streamlit.app/`) for visualization, exploration, and direct download of generated datasets.

We envision GraphUniverse as a versatile tool for diverse research applications, from targeted generalization benchmarks to large-scale data generation and augmentation for model pre-training. While our experiments demonstrate its immediate utility, they represent only a fraction of what is possible with controllable graph family generation. Thus, we release GraphUniverse to the community as a flexible framework, inviting extensions and adaptations to explore new frontiers in graph learning.

## 2 RELATED WORK

The evaluation of graph learning models has evolved from early, limited-scope benchmarks (Dwivedi et al., 2023; Morris et al., 2020) to large-scale, real-world datasets. The Open Graph Benchmark (OGB) (Hu et al., 2020) was a significant step forward, providing standardized protocols on large graphs that revealed critical challenges in generalization. Subsequently, the GOOD benchmark (Gui et al., 2022) introduced a focus on out-of-distribution (OOD) generalization by creating splits designed to test robustness to covariate and concept shifts. However, a fundamental limitation of these real-world benchmarks is their static nature. The datasets are fixed, the properties of the data splits are not tunable, and as recent critiques have noted, they often lack sufficient coverage of important graph properties like heterophily, limiting their utility for systematic model analysis (Bechler-Speicher et al., 2025).

Recognizing the limitations of static datasets, a growing consensus advocates for high-fidelity synthetic data generation as a path toward more principled and scalable evaluation (Bechler-Speicher et al., 2025; Wang et al., 2025). The most prominent effort in this direction is GraphWorld (Palowitch et al., 2022), which uses synthetic generation to study model performance across a space of graph properties. Related efforts like CGT (Zahirnia et al., 2023) and the metadata-driven approach (Li et al., 2023) provide valuable insights into model behavior by mapping real-world graphs to synthetic equivalents and analyzing performance across graph properties, respectively. While these approaches

enable controlled analysis, they remain confined to generating independent, single graphs. This restricts evaluation to the transductive setting, where models are tested on the same graph structure seen during training, and fundamentally prevents the study of a model's ability to generalize to entirely new and unseen graphs.

The importance of synthetic data extends beyond benchmarking to foundation model development. GraphFM (Lachi et al., 2024) leverages GraphWorld-style synthetic graphs to expand its pretraining corpus, though limited to transductive settings, while OpenGraph (Xia et al., 2024) uses LLMs to augment existing datasets—mirroring synthetic data's established role in computer vision (Tobin et al., 2017) and NLP (Wang et al., 2023). However, existing graph generation frameworks cannot provide the inductive generalization and systematic coverage of graph modalities that robust foundation models require (Bechler-Speicher et al., 2025; Wang et al., 2025)—a capability limited to multi-graph approaches.

GraphUniverse directly addresses these limitations. Unlike static benchmarks, it provides a generative framework capable of producing unlimited data with fine-grained control over structural properties. Critically, unlike GraphWorld and other single-graph approaches, it generates entire families of graphs with shared semantic meaning, enabling systematic study of inductive generalization. Furthermore, our experiments validate that GraphUniverse-generated data closely mirrors real-world model behavior, suggesting its potential as a complementary data source for model development, including pre-training applications. We present GraphUniverse as an open-source tool that researchers can extend and adapt for their specific needs, whether for controlled benchmarking or as a basis for more sophisticated data augmentation strategies for Graph Foundation Model developement, of which a detailed discussion is provided in Appendix B.

## 3 BACKGROUND

Community-based graph generation provides interpretable control over node-level properties and their relationships, naturally supports community detection tasks, and reflects the modular organization commonly observed in real-world networks (Fortunato, 2010). This section revisits some previous works on this topic that GraphUniverse draws inspiration from.

---

Let $G = (V, E)$ be an undirected graph with $|V| = n$ nodes and $A \in \{0, 1\}^{n \times n}$ its adjacency matrix, where $A_{ij} = A_{ji}$ and $A_{ii} = 0$. Let $k \in \mathbb{N}$ be the number of communities and $b_i \in \{1, \ldots, k\}$ the community label of node $i$. We denote by $P \in [0, 1]^{k \times k}$ the (symmetric) block/community edge probability matrix with entries $P_{rs}$, with $r, s \in \{1, \ldots, k\}$.

---

**Stochastic Block Model (SBM).** SBMs (Holland et al., 1983) generates a graph by first uniformly sampling labels $b_1, \ldots, b_n$, then drawing edges independently as

$$A_{ij} \sim \text{Bernoulli}(P_{b_i b_j}) \quad (1 \leq i < j \leq n).$$

SBM is used in two complementary ways; (i) the *inference view*: given a single observed $A$, estimate $(b, P)$; and (ii) the *generative view* (the one adopted in this work): given $(n, P)$, sample $A$.

**Degree-Corrected SBM (DC-SBM).** A limitation of SBM is that it enforces homogeneous expected degrees within a community, since edge probabilities depend only on block membership. To address this, the original DC-SBM (Karrer & Newman, 2011) implementation introduces node-specific degree factors $\theta_i > 0$ and a nonnegative block matrix $\Lambda \in \mathbb{R}_{\geq 0}^{k \times k}$. Moreover, they shift focus from simple graphs to *multigraphs*, i.e. a graph with multi-edges and self-loops. In its original *Poisson multigraph* form, edges are counts with rates:

$$A_{ij} \sim \text{Poisson}(\lambda_{ij}) \ (i \neq j), \quad \frac{A_{ii}}{2} \sim \text{Poisson}\left(\frac{1}{2} \theta_i^2 \Lambda_{b_i b_i}\right), \quad \text{where} \ \lambda_{ij} := \theta_i \theta_j \Lambda_{b_i b_j}.$$

Imposing a per-group *sum-to-one* normalization on degree factors, i.e. $\sum_{i: b_i = r} \theta_i = 1 \ \forall r \in \{1, \ldots, k\}$, the expected number of edges between communities $r$ and $s$ (counting each undirected edge once) becomes:

$$\mathbb{E}[M_{rs}] = \begin{cases} \Lambda_{rs}, & r \neq s, \\ \frac{1}{2} \Lambda_{rr}, & r = s, \end{cases}$$

so $\Lambda$ controls the total number of edges in a block, while $\theta$ redistributes degrees within blocks. This Poisson formulation is standard in inference-focused works (Karrer & Newman, 2011; Abbe, 2018).

**Generative Bernoulli Formulation.** Since the original DC-SBM naturally generate Poisson multi-graphs, a common approach to get simple graphs from them is to collapse multi-edges into single undirected edges and remove self-edges after generation (as done in GraphWorld (Palowitch et al., 2022)). However, this leads to a systematic but unpredictable mismatch between the input parameters and the properties of the output graph. Therefore, we choose to work directly with a Bernoulli reformulation of the DC-SBM algorithm as the basis for our generator (Rohe et al., 2018). Its justification and precise correspondence with the Poisson DC-SBM are deferred to Appendix C. Moreover, a discussion of the limitations of relying on DC-SBM models (e.g. lack of higher order structures) is provided in Appendix D.

## 4 GRAPHUNIVERSE

While DC-SBMs generate individual graphs with controllable community structure, they cannot support inductive generalization studies due to independent graph generation with weak community-specific signals. To address this, we introduce a hierarchical generation framework that decouples global community properties from local graph characteristics, enabling systematic investigation of model performance across semantically related graph distributions.

### 4.1 THREE-LEVEL ARCHITECTURE

Our framework is organized hierarchically into three levels, each controlling different aspects of graph generation—see Figure 1.

**Universe Level (Global Community Properties).** At the top level, a *Graph Universe* (left panel in Figure 1) defines a master set of $K$ persistent communities. These communities, assigned at the node-level, retain stable semantic identities across all generated graphs, specified by:

- *Structural patterns*: The universe-level edge propensity matrix $\tilde{\mathbf{P}} \in \mathbb{R}^{K \times K}$ encodes relative inter-community connection strengths. Unlike standard DC-SBMs with uniform block probabilities, we introduce heterogeneity by generating $\tilde{P}_{rs} = 1 + \xi_{rs}$ where $\xi_{rs} \sim \mathcal{N}(0, (2\epsilon)^2)$ with variance parameter $\epsilon$. This perturbation is symmetrized and rescaled to preserve target homophily and degree constraints (details in App. E), yielding fine-grained structural variation across community pairs.

- *Degree profiles*: A community-specific degree propensity vector $\boldsymbol{\delta} = (\delta_1, \dots, \delta_K) \in [-1, 1]^K$, with $\delta_k \sim \text{Uniform}(-1, 1)$, determines the characteristic degree propensity of each community. Unlike standard DC-SBMs where degree factors are independent of community membership, each $\delta_k$ anchors community $k$ in the degree spectrum, with $\delta_k = -1$ corresponding to low-degree node tendencies and $\delta_k = +1$ to high-degree ones.

- *Feature distributions*: Community centroids $\boldsymbol{\mu}_k \in \mathbb{R}^d$, for $k \in \{1, \dots, K\}$, are sampled from $\boldsymbol{\mu}_k \sim \mathcal{N}(\mathbf{0}, \sigma_{\text{center}}^2 \mathbf{I}_d)$, with $\sigma_{\text{center}}^2$ controlling between-community feature separation. Node features within each community $k$ are then generated from $\mathcal{N}(\boldsymbol{\mu}_k, \sigma_{\text{cluster}}^2 \mathbf{I}_d)$, where $\sigma_{\text{cluster}}^2$ controls within-community feature variance.

**Family Level (Generation Constraints).** A *Graph Family* (middle panel in Figure 1) specifies allowed ranges for graph-level parameters while maintaining consistency through the community-behaviour defined by the universe-level identities:

- *Structural ranges*: Target homophily $h \in [h_{\min}, h_{\max}]$ and average degree $d \in [d_{\min}, d_{\max}]$.
- *Size constraints*: Number of nodes $n \in [n_{\min}, n_{\max}]$ and number of participating communities $k \in [k_{\min}, k_{\max}]$ per graph, with $k_{\max} \leq K$.
- *Coupling parameters*: Degree separation $\rho \in [\rho_{\min}, \rho_{\max}]$ controls the overlap between community degree distributions (low $\rho$ yields broad overlap, high $\rho$ yields well-separated distributions), and degree distribution parameters such as power-law exponent $\alpha \in [\alpha_{\min}, \alpha_{\max}]$.

**Graph Level (Instance Generation).** Individual graphs are generated as *Graph Sample* instances (right panel in Fig. 1), each obtained by sampling specific values from family-level ranges and inheriting community properties from the universe. The full procedure is described in next section.

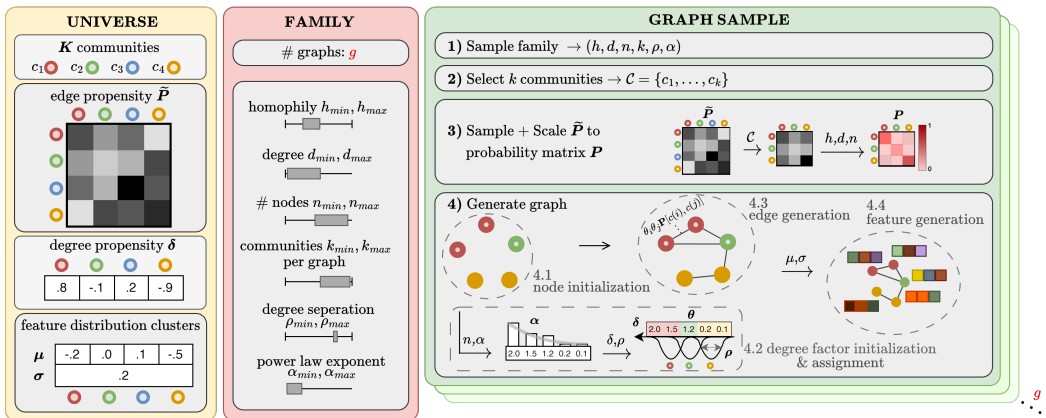

Figure 1: Overview of GraphUniverse generation methodology.

## 4.2 GRAPH INSTANCE GENERATION PROCESS

Each graph instance is generated in four phases, as shown in the *Generate Graph* section in Figure 1:

*Phase 1: Parameter Sampling.* Graph-specific parameters are drawn uniformly from family ranges:

$$(n, k, h, d, \rho, \alpha) \sim \text{Uniform}\Big( [n_{\min}, n_{\max}] \times \cdots \times [\alpha_{\min}, \alpha_{\max}] \Big). \quad (1)$$

*Phase 2: Community Selection.* We randomly select $k$ communities $\mathcal{C} = \{c_1, \ldots, c_k\} \subseteq \{1, \ldots, K\}$ from the universe to appear in this particular graph.

*Phase 3: Probability Matrix Construction.* We extract the $k \times k$ submatrix

$$\mathbf{P}_{\text{sub}}[i, j] \ = \ \mathbf{P}[c_i, c_j], \qquad i, j \in \{1, \ldots, k\}.$$

To obtain valid Bernoulli probabilities with the sampled graph-level properties, we rescale in two stages: (i) *homophily adjustment*: apply separate scaling factors to diagonal vs. off-diagonal entries so that the within- and between-community ratio matches the target homophily $h$; (ii) *density adjustment*: apply a global multiplier so that the mean entry matches the target edge density $d/(n-1)$ for $n$ nodes.

The resulting matrix $\mathbf{P}_{\text{scaled}}$ preserves the heterogeneity of $\mathbf{P}_{\text{sub}}$ while satisfying both constraints (see Appendix E).

*Phase 4: Graph Realization with Community Properties.* The final graph is generated as follows:

1. *Node assignment.* Nodes are distributed uniformly across the selected communities $\mathcal{C}$.
2. *Degree factors.* For each node $i$ in community $c(i)$, we assign a degree factor $\theta_i$ by coupling degree distributions to communities. We first sample power-law degree factors and sort them as $(\theta_{(1)}, \ldots, \theta_{(n)})$. Each community's degree center $\delta_{c(i)}$ maps to a preferred rank $\mu_{c(i)} = \frac{1 + \delta_{c(i)}}{2}(n-1)$, and we assign degree factors by sampling rank indices from $\mathcal{N}(\mu_{c(i)}, \sigma^2)$ truncated to $[1, n]$, where $\sigma^2$ is determined by the degree separation parameter $\rho$. Full details are in Appendix F.
3. *Edge generation.* Each pair $(i, j)$ with $i < j$ is connected independently with probability

$$P_{ij} \ = \ \min\big(1, \ \theta_i \theta_j \mathbf{P}_{\text{scaled}}[c(i), c(j)]\big). \quad (2)$$

   After sampling, we verify connectivity and connect any disconnected components by adding edges that minimize deviation from the target block structure $\mathbf{P}_{\text{scaled}}$ (details in Appendix G).
4. *Feature generation.* Finally, node features are sampled from community-specific Gaussian distributions:

$$\mathbf{x}_i \sim \mathcal{N}(\boldsymbol{\mu}_{c(i)}, \ \sigma^2 \mathbf{I}). \quad (3)$$

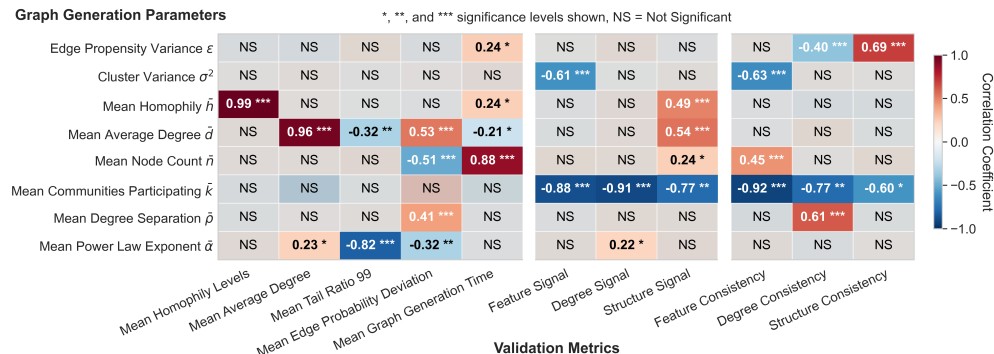

Figure 2: Parameter sensitivity heatmap from 100 randomized graph families with all parameters simultaneously varied across complete ranges. Pearson correlation coefficients are shown with stars indicating significance levels. NS indicates no statistically significant correlation.

## 4.3 VALIDATION OF GRAPHUNIVERSE

To ensure our multiple graph generation framework produces high-fidelity graphs with the intended properties and learnable signals within and across graphs, we conduct a comprehensive validation study into three critical aspects: graph properties, signal strength, and cross-graph consistency.

**Validation Metrics.** We define three categories of validation metrics. *Graph Property* metrics verify that generated graphs match target structural characteristics, including homophily levels, average degree, degree distribution tails, and deviations from expected community edge probability matrices. *Signal Strength* metrics assess the predictability of community labels using different node-level features (node features, degree, and multi-hop neighborhood structure), ensuring graphs contain learnable signals for downstream tasks. *Cross-Graph Consistency* metrics evaluate whether community identities remain semantically consistent across different graph instances through feature centroid similarity, structural pattern correlation, and degree ranking preservation. Detailed definitions for all metrics are provided in Appendix I.

**Parameter Sensitivity Analysis.** We generate 100 distinct graph families—30 graphs each—with completely randomized parameter configurations sampled uniformly across broad ranges (further details in Appendix H). This stress-tests the framework's robustness by capturing parameter interactions at extreme values rather than nominal operating points. We assess parameter-metric relationships using Pearson correlations on family-level means, reporting only statistically significant correlations ($p < 0.05$). The correlation heatmap of Figure 2 shows the parameter (y-axis) responsiveness across all validation metrics (x-axis).

**Validation Results.** Graph property metrics (left panel in 2) show expected strong correlations: parameters precisely control homophily and average degree, with slight deviations under extreme settings reflecting our multiplicative edge generation process. Signal strength metrics (middle panel in 2) confirm theoretical expectations: cluster variance controls feature signals, homophily/degree enhance structure signals, and fewer communities simplify classification. Cross-graph consistency metrics (right panel in 2) provide the strongest validation of our hierarchical design, with propensity variance governing structure consistency, degree separation controlling degree consistency, and cluster variance determining feature consistency. A detailed analysis of the parameter validation results, as well as individual parameter effect plots, are provided in Appendix Section J.

## 4.4 SCALABILITY

GraphUniverse demonstrates linear scaling across graph sizes, enabling efficient large-scale evaluation. Table 1 shows generation performance measured on an AMD Ryzen 7 5700U CPU processor (single-threaded), averaged over 100 graphs.

Table 1: GraphUniverse generation stats.

| Avg. number of nodes | Time per graph (sec) | Throughput (graphs/sec) |
|---|---|---|
| 10 | **0.002** | 449.7 |
| 100 | **0.023** | 42.8 |
| 500 | **0.349** | 2.9 |
| 1000 | **1.309** | 0.8 |

## 4.5 INTERACTIVE EXPLORATION TOOL

To complement the open-source package, we provide an interactive Streamlit application for code-free exploration of the GraphUniverse generator, available both as a hosted demo (`https://graphuniverse.streamlit.app/`) and for local deployment via the PyPI package. The app allows users to define a generation universe, tune family parameters (e.g., homophily, degree distribution), and instantly visualize the resulting graphs, their properties and validation metrics. Generated graph families can be downloaded as PyTorch Geometric `InMemoryDataset` objects.

## 5 BENCHMARKING

The ability to generate diverse graph families with fine-grained control over their properties opens up countless avenues for systematic model evaluation. To demonstrate this potential, we present a benchmarking suite designed to probe three fundamental research questions (RQ) of inductive performance, generalization and robustness in modern GNNs. This investigation, while comprehensive, represents just one of the many possible explorations that GraphUniverse makes possible.

### 5.1 EXPERIMENTAL SETUP

**Implementation.** All experiments are conducted using the TopoBench benchmarking framework (Telyatnikov et al., 2025), which we extend with a custom GraphUniverse loader to systematically define and iterate over graph generation parameters. Across our benchmarking, we evaluate models in both inductive and transductive settings on two distinct tasks: node-level community detection (classification) and graph-level triangle counting (regression). Unless otherwise specified, we consider a set of fixed dataset generation parameters—such as the number of graphs in inductive settings (1000), or the number of nodes in transductive ones (1000)—to ensure a consistent baseline across experiments (defaults lists in Appendix M.1).

**Models.** We evaluate a diverse set of architectures representing major paradigms in contemporary graph learning (a brief description of each of them can be found in Appendix M.2): DeepSet (Zaheer et al., 2017), GraphMLP (Hu et al., 2021), GCN (Kipf & Welling, 2017), GraphSAGE (Hamilton et al., 2017), GIN (Xu et al., 2019), GATv2 (Brody et al., 2022), TopoTune (Papillon et al., 2025), Neural Sheaf Diffusion (Bodnar et al., 2022), and GPS (Rampášek et al., 2022) (see Appendix M.2).

**Hyperparameter Optimization.** For each model-dataset configuration, we conduct comprehensive grid search hyperparameter optimization using architecture-specific parameter grids detailed in the Appendix M.3. Each configuration is evaluated across 3 different dataset instantiations generated with the same input parameters but different data random seeds, with the configuration achieving the highest mean validation performance selected for final test evaluation.

**Evaluation Metrics.** We report test accuracy and mean absolute error (MAE) for community detection and triangle counting tasks, respectively (averaged across 3 random data seeds with standard deviations).[1] For both inductive and transductive experiments, we consider a 70/15/15 training/val/test split.

### 5.2 RQ1: DO GRAPH LEARNING MODELS PERFORM DIFFERENTLY IN THE INDUCTIVE SETTING, IN FUNCTION OF KEY VARIED GRAPH PROPERTIES?

**Motivation.** Existing synthetic graph generation models (GraphWorld (Palowitch et al., 2022)) only allow for benchmarking models on single graphs in transductive settings, but real applications require generalization to unseen graphs with different properties.

**Experimental Design.** We systematically vary three fundamental graph properties across families of 1000 graphs each: homophily range ($[0.0, 0.1]$, $[0.4, 0.6]$, $[0.9, 1.0]$), average degree range ($[1, 5]$, $[5, 10]$, $[10, 20]$), and cluster variance (basically feature noise, setting as options 0.2, 0.5 and 0.8), keeping all other graph family parameters at default values. We directly compare these inductive

---

[1]Accuracy is appropriate given uniform community size distributions enforced across all graph families.

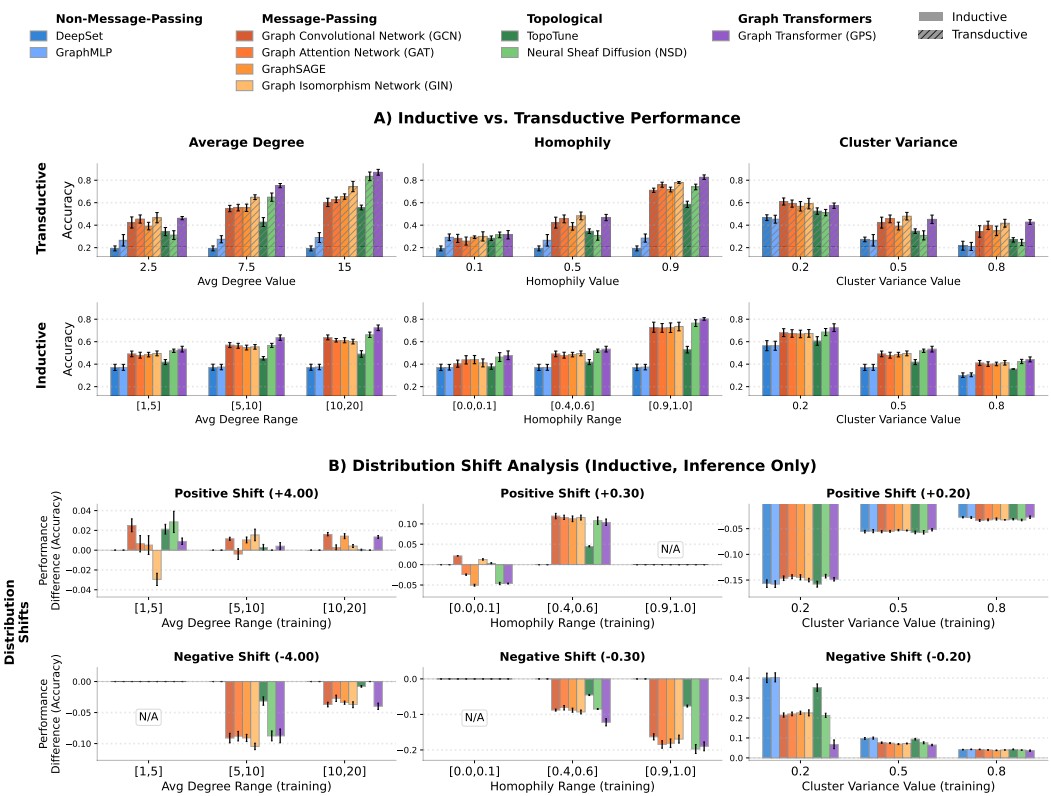

Figure 3: A) Inductive (graph families of 1000 graphs) versus transductive (single graphs) test accuracy on community detection across different graph properties, with each architecture individually optimized. B) Distribution shift analysis: best-performing inductive models evaluated on graph families with shifted properties from the same Universe. Plots show accuracy changes under distributional shifts, with x-axis indicating the original training domain. N/A indicates shifts beyond feasible parameter bounds.

results against equivalent single-graph transductive evaluation using GraphWorld-style generation (with mean property of corresponding inductive setting). Figure 3.A reports on the results. Appendix L extends the homophily analysis to specific heterophilic GNN architectures.

**Insight 1: Distinct Model Ranking Profiles Across Settings.**   We observe a striking divergence in model performance rankings between the inductive and transductive settings (Fig. 3.A). While GPS and non-message passing architectures (Deepset, GraphMLP) consistently achieve top and bottom performances, respectively, other architectures show clear setting-dependent strengths. For example, Neural Sheaf Diffusion excels inductively but falters transductively, suggesting its topological biases aid generalization across graphs. Conversely, GIN dominates transductively but fails inductively, indicating its success may stem from memorizing a single graph's structure. These shifts reveal that transductive performance is not always a reliable proxy for a model's ability to generalize.

**Insight 2: Transductive Setting Amplifies Graph Property Effects**   While increasing graph homophily and average degree improves performance in both settings, these benefits are significantly amplified in the transductive paradigm. As seen in Figure 3.A, the performance gap between low and medium homophily or degree configurations is far more pronounced transductively. This suggests that when models have access to the entire graph structure during training, they can better exploit favorable properties. Consequently, transductive evaluation may overestimate a model's true sensitivity to these structural characteristics.

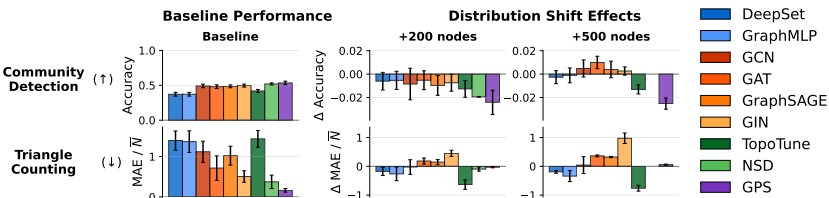

Figure 4: Left: baseline accuracy on original graphs. Right: performance changes ($\triangle$) when evaluating on larger graphs (+200, +500 nodes). Triangle counting uses normalized MAE by average graph size $\overline{N}$. Out-of-memory error for NSD in largest graphs (+500).

## 5.3 RQ2: How robust are graph learning models under distribution shifts?

**Motivation.** Deployed models must handle property distribution shifts between training and test data, where target graphs exhibit different structural properties than those seen during training. Understanding how performance degrades under controlled property shifts is crucial for graph foundation model development.

**Experimental Design.** Using optimal model configurations identified in the inductive experiments of RQ1, we evaluate performance degradation under controlled shifts. That is, for each baseline property setting (homophily, average degree, cluster variance), we generate, using the same Universe, a new test family with systematic shifts: $\pm 0.1$ homophily, $\pm 4$ average degree, and $\pm 200$ nodes per graph. The optimal models from RQ1 are evaluated on the shifted families. This provides a systematic characterization of each architecture's sensitivity to different types of distribution shifts.

**Key Insight: Model Robustness is Context-Dependent, Not Universal.** Our experiments (Fig. 3.B) reveal that model robustness is not an intrinsic property but emerges from specific interactions between a model's architecture and the graph's properties. We found that identical distributional shifts can produce opposite effects depending on the training regime; for example, increasing homophily can harm a model's performance in a low-homophily setting but improve it in a medium one. This exposes architecture-specific vulnerabilities: GIN proves highly sensitive to degree shifts in low-degree graphs, while models like GraphSAGE show a counterintuitive performance drop when homophily is increased from a low baseline. These findings suggest that models often achieve high performance through narrow specialization on training regimes rather than robust generalization, highlighting the critical need for training data diversity.

## 5.4 RQ3: Do models trained on small graphs generalize to bigger graphs?

**Motivation.** Real-world deployment often requires models trained on smaller graphs to handle larger instances. Understanding size generalization is critical for practical scalability of graph learning models. While we focused up until now on node-level community detection, we also evaluate triangle counting as a structural graph-level task, though our framework supports other graph-level tasks.

**Experimental Design.** We evaluate generalization across graph sizes using two complementary tasks. For community detection (node-level, local task), we train on graphs ranging from 50 to 200 nodes, then evaluate on families (from the same Universe) with 250-400 and 550-700 nodes. For triangle counting (graph-level, global task), we follow the same size progression.

**Key Insight: Graph-level MPNNs Fail to Generalize to Larger Graphs.** We can see (Fig. 4) that node-level tasks (community detection) show minimal sensitivity to graph size (2% degradation) due to local neighborhood aggregation, except for GPS and NSD which suffer minor drops from their more global components (positional encodings, attention). Graph-level tasks (triangle counting) initially show only GPS, NSD, and GIN effectively solving the task, but while GPS and NSD maintain performance when scaling to larger graphs, GIN fails to generalize, suggesting traditional MPNNs overfit to training graph sizes.

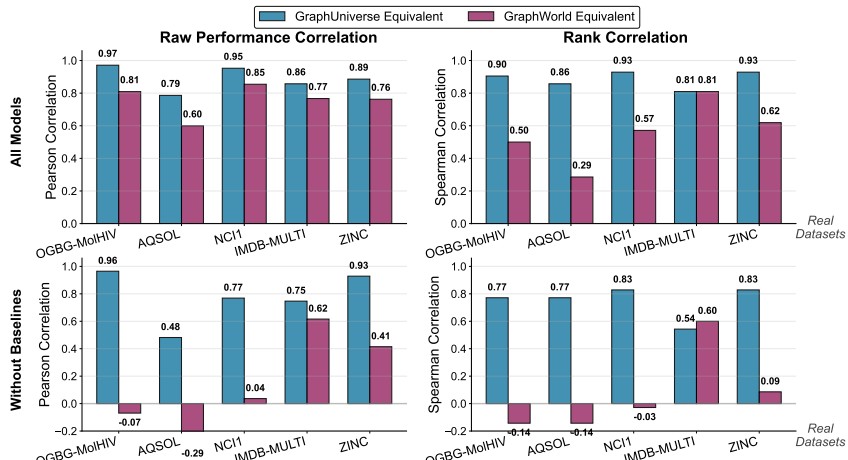

Figure 5: Model ranking correlations between real datasets and equivalent synthetic datasets. Rankings computed via bootstrap analysis. GraphUniverse (blue) shows consistently higher alignment with real-world model rankings compared to GraphWorld (purple) across both raw performance and rank-based metrics. "Without Baselines" excludes DeepSet and GraphMLP to avoid overestimation.

## 5.5 RQ4: DOES GRAPHUNIVERSE ACCURATELY PREDICT REAL-WORLD MODEL PERFORMANCE?

**Motivation.** A critical question for any synthetic benchmark is whether insights gained extend to real-world performance. We demonstrate that GraphUniverse effectively predicts model behavior on real, inductive datasets, providing stronger alignment than single-graph approaches.

**Experimental Design.** For five real-world inductive datasets we extract key structural properties and generate two synthetic equivalents: (1) a **GraphUniverse equivalent** matching property distributions (5th-95th percentiles), and (2) a **GraphWorld equivalent** using mean values in a single graph. We map dataset-specific communities to enable fair comparison. We train models on original tasks for real-world datasets and community detection for synthetic datasets. After optimizing the same suite of models as in previous RQs, we compute model rankings via bootstrap analysis and assess correlation between mean synthetic and real rankings. Full extraction details are in Appendix K.1.

**Key Insight: GraphUniverse Provides Superior Real-World Alignment.** Figure 5 shows GraphUniverse achieves substantially higher correlations with real datasets than GraphWorld. We analyze both all models and graph-aware models separately, as non-message-passing baselines (DeepSet, GraphMLP) consistently underperform across all settings, artificially inflating correlations. Focusing on message-passing models reveals that GraphUniverse shows positive correlations for all datasets while GraphWorld shows negative correlations for half. Model-level analysis in Figures 10 and 11 of Appendix K.2 confirms that GraphUniverse accurately captures how model rankings vary across datasets with different structural properties. These results validate GraphUniverse as a meaningful proxy for real-world evaluation, particularly for rapid prototyping and systematic studies.

## 6 CONCLUDING REMARKS

We introduce GraphUniverse, a synthetic graph generation framework designed to address a critical gap in graph learning: the systematic evaluation of inductive generalization. By generating graph families with consistent semantics and tunable structural and feature properties, GraphUniverse provides the scalable, controlled data required to rigorously assess a model's ability to generalize across diverse and unseen graphs. Our experiments reveal valuable insights into model robustness and generalization. We release GraphUniverse as an open-source tool to unlock new research directions in the principled development and validation of both existing and novel architectures —including potential applications in graph foundation model research, though such extensions would require additional development beyond our current framework.

## REPRODUCIBILITY STATEMENT

GraphUniverse is available as a Python package at `https://pypi.org/project/graph-universe/`, with full source code at `https://github.com/LouisVanLangendonck/GraphUniverse`. All experiments can be reproduced using the TopoBench framework Telyatnikov et al. (2025).

## ACKNOWLEDGMENTS

This work was supported by Grant PCI2023-145974-2 funded by MI-CIU/AEI/10.13039/501100011033 and cofunded by the European Union (GRAPHS4SEC project). This work is also supported by the Catalan Institution for Research and Advanced Studies (ICREA Academia). Louis Van Langendonck is supported by the predoctoral program AGAUR-FI ajuts (2025 FI-3 00065) Joan Oró, of the Department of Research and Universities of the Generalitat of Catalonia, as well as the European Social Plus Fund. Nina Miolane acknowledges the NSF grant 2313150 and the NSF CAREER Award 240158. Guillermo Bernardez Gil acknowledges funding from Chan Zuckerberg Initiative, UC Noyce Foundation, and Arlequin AI.

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

## A    USE OF LARGE LANGUAGE MODELS (LLMS) IN PAPER WRITING

We drafted all content ourselves and then used LLMs to improve grammar, rephrase text, and shorten extensive sections. The models are used as editorial tools to help make our writing clearer and more concise.

## B    DISCUSSION ON ROLE AND POTENTIAL IMPACT IN GRAPH FOUNDATION MODEL DEVELOPMENT

While our primary contribution focuses on systematic inductive evaluation, GraphUniverse has potential relevance to graph foundation model (GFM) development. This section outlines two potential applications.

### B.1    SYSTEMATIC EVALUATION TESTBED

GraphUniverse enables rapid generation of diverse graph families for comprehensive GFM assessment across controlled distribution shifts. Unlike static benchmarks, our framework allows systematic robustness evaluation by generating unlimited scenarios with known variations in structural properties (homophily, density, degree distributions). This provides a precise, low-cost method for identifying architectural vulnerabilities and understanding failure modes, directly addressing concerns about current evaluation practices raised in recent position papers.

### B.2    DATA AUGMENTATION POTENTIAL

Our framework could potentially be extended to serve as a sophisticated data augmentation tool for GFM pre-training. By fitting GraphUniverse parameters to match statistical properties of target domains, researchers could generate novel but realistic graph instances that preserve essential structural relationships. This aligns with successful synthetic data strategies in computer vision and NLP, where augmentation has proven valuable for improving model robustness Tobin et al. (2017); Wang et al. (2023).

The key strength of this approach would be leveraging our framework's flexibility to generate datasets covering data modalities completely unseen in original training sets. By systematically varying homophily levels, density, and graph sizes beyond those present in real datasets, researchers could potentially reduce overfitting to specific dataset characteristics and train more general models capable of robust performance across diverse graph types.

### B.3    LIMITATIONS AND FUTURE DIRECTIONS

These applications represent potential future directions rather than validated capabilities, with several concrete challenges requiring resolution. First, identifying meaningful 'communities'—the building blocks of our framework—in real-world datasets presents domain-specific challenges. While some datasets naturally provide community structure (e.g., atom types in molecular data), others would require sophisticated clustering approaches, potentially incorporating positional information or employing a two-stage process: initial unlabeled pre-training followed by clustering on learned embeddings to fit the data generator.

Second, realistic feature generation would require careful fitting to real data distributions, most straightforwardly achieved by computing per-community, per-dimension statistics (means and standard deviations) and sampling accordingly. Our current DC-SBM foundation may require extension to more sophisticated generators to fully capture the complexity needed for realistic augmentation across diverse domains.

Nevertheless, recent GFM surveys explicitly call for these types of evaluation and generation capabilities, suggesting clear alignment with community needs and representing a promising line of future work Wang et al. (2025); Bechler-Speicher et al. (2025).

## C  BERNOULLLI FORMULATION OF SIMPLE DC-SBMS

**Poisson multigraph.**  In the classical DC-SBM (Karrer & Newman, 2011), edges are Poisson counts

$$A_{ij} \sim \text{Poisson}(\lambda_{ij}), \qquad \lambda_{ij} = \theta_i \theta_j \Lambda_{b_i b_j}, \tag{4}$$

with per-community normalization $\sum_{i \in r} \theta_i = 1$. The expected block edge totals are then

$$\mathbb{E}[M_{rs}] = \begin{cases} \Lambda_{rs}, & r \neq s, \\ \frac{1}{2}\Lambda_{rr}, & r = s. \end{cases} \tag{5}$$

**Collapsed Poisson simple graph.**  If we form a simple graph by collapsing multi-edges,

$$\widetilde{A}_{ij} = \mathbf{1}\{A_{ij} \geq 1\}, \tag{6}$$

then

$$\Pr[\widetilde{A}_{ij} = 1] = 1 - e^{-\lambda_{ij}}. \tag{7}$$

Since $1 - e^{-x} < x$ for $x > 0$, the collapsed model systematically underestimates edge probabilities and thus block totals, except in the extremely sparse regime where $1 - e^{-\lambda_{ij}} \approx \lambda_{ij}$.

**Bernoulli simple graph.**  Following the approach of Rohe et al. (2018), we can define edges directly as Bernoulli trials,

$$A_{ij} \sim \text{Bernoulli}\left(\min\left(1, \theta_i \theta_j P_{b_i b_j}\right)\right), \tag{8}$$

with per-community mean-one normalization

$$\frac{1}{|V_r|} \sum_{i \in r} \theta_i = 1. \tag{9}$$

The expected block edge totals are then:

- For $r \neq s$: $\mathbb{E}[M_{rs}] = P_{rs} \sum_{i \in r} \sum_{j \in s} \theta_i \theta_j$
- For $r = s$: $\mathbb{E}[M_{rr}] = \frac{1}{2} P_{rr} \sum_{i \in r} \sum_{j \in s, j \neq i} \theta_i \theta_j$ (excluding self-loops)

Under the normalization constraint, these simplify to:

- For $r \neq s$: $\mathbb{E}[M_{rs}] = P_{rs} |V_r| |V_s|$
- For $r = s$: $\mathbb{E}[M_{rr}] = \frac{1}{2} P_{rr} |V_r| (|V_r| - 1)$

**Equivalence.**  To match the Poisson multigraph block expectations, we set:

$$P_{rs} = \frac{\Lambda_{rs}}{|V_r||V_s|} \quad (r \neq s), \qquad P_{rr} = \frac{\Lambda_{rr}}{|V_r|(|V_r| - 1)} \quad (r = s). \tag{10}$$

However, as noted by Rohe et al. (2018), this equivalence only holds when the resulting probabilities satisfy $\theta_i \theta_j P_{b_i b_j} \leq 1$ for all $i, j$. When this constraint is violated, we apply clipping to ensure valid Bernoulli probabilities:

$$\text{edge probability} = \min(1, \theta_i \theta_j P_{b_i b_j}), \tag{11}$$

which introduces a systematic deviation from the Poisson block structure in dense regimes.

**Theoretical justification.**  The theoretical foundation for this approach follows directly from Theorem 3 in Rohe et al. (2018). Their result shows that in sparse regimes where $\lambda_{ij} = O(\alpha_n/n)$ for some sequence $\alpha_n$, there exists a coupling between the thresholded Poisson graph and the direct Bernoulli graph such that

$$\frac{\mathbb{E}\|t(\tilde{A}) - B\|_F^2}{\mathbb{E}\|B\|_F^2} = O(\alpha_n/n), \tag{12}$$

where $t(\tilde{A})$ represents the thresholded Poisson graph and $B$ represents the direct Bernoulli graph. This establishes that the two approaches are asymptotically equivalent in sparse settings.

**Summary.** The Bernoulli DC-SBM with clipping preserves the interpretability of edge probabilities and avoids the systematic underestimation of the collapsed Poisson approach, while providing exact control over the simple graph structure. The theoretical equivalence established by Rohe et al. (2018) validates this approach in sparse regimes, while the controlled deviation from Poisson block expectations due to clipping represents a principled trade-off that enables direct generation of simple graphs with desired structural properties.

# D DISCUSSION ON THE LIMITATIONS OF DEGREE-CORRECTED STOCHASTIC BLOCK MODELS AS DATA GENERATOR

The DC-SBM formulation underlying GraphUniverse carries inherent limitations that merit explicit discussion. We categorize these limitations into two types, each with different implications for our framework's applicability and future development.

## D.1 READILY EXTENSIBLE LIMITATIONS

Several limitations stem from design choices made for experimental simplicity and interpretability. Features such as deterministic community membership, discrete non-overlapping communities, and uniform community size distributions could readily be implemented as extensions to our current framework. We deliberately chose these simplifications to maintain clear experimental control and focus on core structural phenomena that are easily identifiable, controllable, and translatable to real-world settings.

For researchers interested in studying specific phenomena like overlapping communities, gradual community transitions, or hierarchical community structures, our framework provides a solid foundation that can be extended while preserving the systematic control that makes synthetic benchmarks scientifically valuable.

### D.1.1 IMPLEMENTING OVERLAPPING COMMUNITIES

To illustrate the extensibility of our framework, we outline how overlapping communities could be implemented through two additional universe-level parameters:

**Co-occurrence Count Distributions**: For each community $k$, define a discrete probability distribution over the number of additional communities a node can belong to. This could be implemented as user-defined distributions or generated using negative binomial distributions with controllable parameters for mixing probability and distribution shape. Setting all distributions to concentrate on zero recovers the current non-overlapping scenario.

**Community Mixing Matrix**: A symmetric $K \times K$ matrix $\mathbf{M}$ where each row sums to 1, controlling how membership is distributed among overlapping communities. Entry $\mathbf{M}_{i,j}$ represents the relative strength of membership in community $j$ when a node's primary assignment is to community $i$.

The extension process would work as follows:

1. Assign each node a primary community as before

2. Sample the number of additional communities from the co-occurrence distribution

3. Randomly select additional communities and use the mixing matrix to determine membership weights

4. Compute final node behavior as weighted combinations—edge probabilities become weighted by both nodes' membership vectors, degree factors are determined by the dominant community membership, and features are drawn as weighted combinations from respective community centroids

Formally, if node $i$ has membership vector $\boldsymbol{\pi}_i \in [0,1]^K$ with $\sum_{k=1}^{K} \pi_{i,k} = 1$, then:

$$P_{ij} = \min\left(1, \theta_i \theta_j \sum_{r=1}^{K} \sum_{s=1}^{K} \pi_{i,r} \pi_{j,s} P_{\text{scaled}}[r,s]\right) \tag{13}$$

$$\theta_i = \theta_{i,k^*} \text{ where } k^* = \arg\max_k \pi_{i,k} \tag{14}$$

$$\mathbf{x}_i \sim \mathcal{N}\left(\sum_{k=1}^{K} \pi_{i,k} \boldsymbol{\mu}_k, \sigma^2 \mathbf{I}\right) \tag{15}$$

This approach preserves all existing framework properties while enabling systematic control over community overlap patterns, demonstrating how our hierarchical design facilitates principled extensions.

### D.2 FUNDAMENTAL LIMITATIONS

More significant limitations arise from our inability to directly control complex motif-driven structures, geometric arrangements, or specific higher-order patterns commonly found in real-world networks. The DC-SBM cannot generate graphs with predetermined triangular motifs, star patterns, or geometric constraints, representing a fundamental constraint on the types of graph structures our framework can produce.

This limitation could potentially bias our evaluation toward models that perform well on community-structured data while potentially penalizing architectures designed for other graph topologies. However, our Research Question 4 (Section 5.5) experiments provide encouraging evidence that despite these structural constraints, GraphUniverse-generated datasets effectively predict real-world model performance across diverse tasks, including molecular property prediction that depends heavily on complex functional groups and higher-order chemical structures (Wu et al., 2023).

### D.3 IMPLICATIONS AND FUTURE DIRECTIONS

The transferability we observe suggests that community-centric evaluation captures sufficient fundamental graph learning capabilities—the interplay between local structure, features, and connectivity patterns—for meaningful model assessment across diverse domains. Nevertheless, extending our framework to incorporate more sophisticated generative models while maintaining systematic experimental control represents a valuable direction for future work, particularly for applications requiring finer control over specific structural motifs or geometric properties.

## E SCALING RAW PROPENSITY TO BERNOULLI PROBABILITY MATRIX WITH DESIRED EXPECTED HOMOPHILY AND AVERAGE DEGREE

To introduce controllable heterogeneity, we generate a raw *propensity matrix* $\widetilde{P} \in \mathbb{R}_{\geq 0}^{k \times k}$ as

$$\widetilde{P}_{rs} = 1 + \xi_{rs}, \qquad \xi_{rs} \sim \mathcal{N}(0, (2\epsilon)^2), \quad \epsilon \in [0,1], \tag{16}$$

where $\epsilon$ controls the variance of the perturbation. Entries are clipped to $[0,2]$ and symmetrized by setting $\widetilde{P}_{rs} = \widetilde{P}_{sr}$. When $\epsilon = 0$ the matrix reduces to the all-ones matrix, while larger values of $\epsilon$ yield increasing heterogeneity across communities.

**Scaling to Target Density and Homophily** The raw propensity matrix $\widetilde{P}$ only specifies relative propensities. Given a graph with $n$ nodes with a uniform distribution of $k$ communities, we want to transform it into a valid probability matrix $P^* \in [0,1]^{k \times k}$ that achieves a user-specified average degree $d$ and homophily level $h$.

Let

$$S_{\text{diag}} = \sum_{r=1}^{k} \widetilde{P}_{rr}, \qquad S_{\text{off}} = \sum_{\substack{r,s=1 \\ r \neq s}}^{k} \widetilde{P}_{rs}.$$

We first apply two scaling factors $\alpha_{\text{diag}}, \alpha_{\text{off}} > 0$ to obtain

$$P'_{rs} = \begin{cases} \alpha_{\text{diag}} \cdot \widetilde{P}_{rr}, & r = s, \\ \alpha_{\text{off}} \cdot \widetilde{P}_{rs}, & r \neq s. \end{cases} \tag{17}$$

The ratio of diagonal to off-diagonal mass is then

$$\frac{\sum_r P'_{rr}}{\sum_{r \neq s} P'_{rs}} = \frac{\alpha_{\text{diag}} S_{\text{diag}}}{\alpha_{\text{off}} S_{\text{off}}}.$$

We now enforce the target homophily constraint by stating enforcing that this ratio equals $h/(1 - h)$, which yields the constraint

$$\frac{\alpha_{\text{diag}}}{\alpha_{\text{off}}} = \frac{h}{1 - h} \cdot \frac{S_{\text{off}}}{S_{\text{diag}}}. \tag{18}$$

Up to this point we are not yet working with actual probabilities so we can scale the diagonal by setting $\alpha_{\text{diag}} == 1$ and calculating $\alpha_{\text{off}}$ by solving equation 18 and scale the off-diagonal by this value.

Next we impose the average degree constraint and scale to obtain actual edge probabilities. Let $n$ be the number of nodes and let the target edge density be

$$\rho_{\text{target}} = \frac{d}{n - 1}. \tag{19}$$

We apply a global scaling factor $\beta > 0$ to obtain the final matrix

$$P^* = \beta P', \tag{20}$$

where $\beta$ is chosen such that the mean entry of $P^*$ equals $\rho_{\text{target}}$, i.e.

$$\beta = \frac{n^2 \rho_{\text{target}}}{\sum_{r,s} P'_{rs}}. \tag{21}$$

Finally, we clip entries of $P^*$ to the interval $[0, 1]$ to ensure valid Bernoulli probabilities.

The resulting matrix $P^*$ satisfies three properties: (i) it preserves the relative heterogeneity induced by $\widetilde{P}$, (ii) it achieves the specified homophily ratio between intra- and inter-community connections, and (iii) it yields an expected average degree of $d$ up to sampling fluctuations. This construction allows graphs to be generated at arbitrary density and homophily levels without discarding the fine-grained structure encoded in $\widetilde{P}$, which is essential for controlled multi-graph family generation.

## F  DETAILS OF COMMUNITY-COUPLED DEGREE FACTORS

For completeness we record the precise definitions used in the degree–community coupling mechanism.

**Overall procedure.**  The coupling process consists of four steps: (1) sample power-law degree factors independently, (2) sort them in ascending order, (3) assign sorted factors to nodes based on community-specific rank sampling, and (4) apply global normalization.

**Power-law degree factor generation.**  We first generate $n$ independent degree factors from a power-law distribution with exponent $\alpha$:

$$\theta_i^{(0)} \sim \text{PowerLaw}(\alpha), \quad i = 1, \ldots, n$$

These are then sorted to obtain the ordered sequence $(\theta_{(1)}, \ldots, \theta_{(n)})$ with $\theta_{(1)} \leq \theta_{(2)} \leq \cdots \leq \theta_{(n)}$.

**Community rank centers.**  Each community $k$ is assigned a degree center $\delta_k \in [-1, 1]$ that maps linearly to a preferred mean rank:

$$\mu_k = \frac{1 + \delta_k}{2}(n - 1)$$

Thus $\delta_k = -1$ corresponds to rank $\mu_k = 0$ (lowest-degree regime), $\delta_k = 0$ to rank $\mu_k = (n - 1)/2$ (middle-degree regime), and $\delta_k = +1$ to rank $\mu_k = n - 1$ (highest-degree regime).

**Rank sampling and assignment.**   For each node $i$ in community $c(i)$, we sample a rank index $\ell_i$ from a truncated Gaussian distribution:

$$\ell_i \sim \mathcal{N}(\mu_{c(i)}, \sigma^2) \text{ truncated to } [1, n]$$

Node $i$ is then assigned degree factor $\theta_i = \theta_{(\ell_i)}$.

**Variance interpolation.**   The sampling variance is set as

$$\sigma^2 = \sigma_{\min}^2 + (1 - \rho)(\sigma_{\max}^2 - \sigma_{\min}^2),$$

where $\rho \in [0, 1]$ is the degree separation parameter. Here $\sigma_{\max} = n$ corresponds to nearly uniform assignments with strong overlap across communities.

**Minimal variance.**   To prevent degenerate overlaps when communities are spread apart in degree space, we set

$$\sigma_{\min} = \max\left(1, \ \min_{k \neq k'} \frac{|\mu_k - \mu_{k'}|}{6}\right),$$

which ensures sufficient separation whenever the community centers $\mu_k$ are far apart.

**Normalization.**   In the classical Bernoulli DC-SBM, degree factors are normalized within each community:

$$\frac{1}{n_r} \sum_{i : b_i = r} \theta_i = 1 \qquad \forall r.$$

In our construction we instead apply a single global normalization

$$\frac{1}{n} \sum_{i=1}^{n} \theta_i = 1,$$

so that the average degree factor is one across all nodes. This choice preserves the relative placement of communities in the degree spectrum, though it does not guarantee exact per-community calibration. Empirically we observe that the global normalization suffices for maintaining the target average degree (see Section 4.3).

## G   CONNECTIVITY CORRECTION ALGORITHM

When the initial edge sampling results in disconnected components, we employ a greedy algorithm to restore connectivity while minimally perturbing the intended block structure. The procedure operates as follows:

**Algorithm Overview:** We iteratively connect the smallest disconnected component to the main graph by selecting edges that best align with the target probability matrix $\mathbf{P}_{\mathrm{sub}}$. After each edge addition, we recompute connected components and repeat until the graph becomes fully connected.

**Connection Selection:** For each potential edge $(i, j)$ between communities $c(i)$ and $c(j)$, we calculate a score based on the current deviation between actual and expected inter-community edge probabilities:

- If the actual probability is below the expected value ($\text{actual}_{c(i),c(j)} < \mathbf{P}_{\mathrm{sub}}[c(i), c(j)]$), adding an edge reduces this negative deviation (preferred option).
- If the actual probability exceeds expectations, we select connections that minimize further deviation.

**Deviation Calculation:** We maintain a normalized actual probability matrix where edge counts are divided by the maximum possible edges between community pairs, then scaled to match the total mass of $\mathbf{P}_{\mathrm{sub}}$ for fair comparison.

This approach optimally balances connectivity requirements with structural fidelity: it improves the match to the target block structure when possible (by connecting under-connected community pairs) and minimizes degradation when connectivity necessitates violating the intended structure, thereby preserving the statistical properties of the generated graph family to the greatest extent possible.

# H  CORE PARAMETER ALLOWED SAMPLING RANGES FOR VALIDATION EXPERIMENT

Table 2: Parameter sampling ranges for randomized validation experiments

| Parameter | Type | Sampling Range |
|---|---|---|
| **Universe Level** | | |
| Edge Propensity Variance ($\epsilon$) | Continuous | [0.0, 1.0] |
| Feature Dimension | Discrete | [10, 100] |
| Center Variance ($\sigma^2_{\text{center}}$) | Continuous | [0.1, 1.0] |
| Cluster Variance ($\sigma^2_{\text{cluster}}$) | Continuous | [0.1, 1.0] |
| **Family Level** | | |
| Min Node Count ($n_{\min}$) | Discrete | [50, 400] |
| Max Node Count ($n_{\max}$) | Discrete | [100, 1000] |
| Min Communities ($k_{\min}$) | Discrete | [2, 15] |
| Max Communities ($k_{\max}$) | Discrete | [4, 15] |
| Homophily Range ($h_{\min}, h_{\max}$) | Range | [0.0, 1.0] |
| Average Degree Range ($d_{\min}, d_{\max}$) | Range | [2.0, 20.0] |
| Degree Separation Range ($\rho_{\min}, \rho_{\max}$) | Range | [0.0, 1.0] |
| Power Law Exponent Range ($\alpha_{\min}, \alpha_{\max}$) | Range | [1.5, 4.5] |

For parameters that represent ranges themselves (e.g., Homophily Range, Average Degree Range), we sample the range bounds from the specified limits and then generate individual range spans with widths between 5% and 20% (randomly drawn) of the parameter space, ensuring meaningful variation while maintaining practical constraints. All experiments use a fixed universe size of $K = 15$ communities. Results shown in Table 2.

Note: For paired parameters (min/max node count and communities), the code ensures logical constraints where maximum values exceed minimum values by appropriate margins.

# I  VALIDATION METRICS IMPLEMENTATION DETAILS

We organize our validation metrics into three categories that capture different aspects of generation quality. An overview table of all validation metrics is given in Table 3:

## I.1  GRAPH PROPERTY METRICS

These metrics verify that generated graphs match their target structural specifications:

**Homophily:** Fraction of edges within communities: $h = \sum_{(i,j) \in E} \mathbb{1}[c(i) = c(j)]/|E|$. This directly validates whether the target homophily level is achieved.

**Average Degree:** Mean node degree across the graph, validating that edge density scaling produces the intended connectivity level.

**Degree Tail Ratio:** Ratio of 99th percentile to mean degree ($\tau_{99} = d_{99}/\bar{d}$), capturing heavy-tailedness of the degree distribution and validating power-law parameter effects.

**Generation Time:** Wall-clock time per graph instance, assessing computational efficiency.

**Mean Probability Matrix Deviation:** This metric quantifies how well the realized graph structure matches the target community connection patterns. For each graph, we compute the deviation between the actual probability matrix $\mathbf{A}_{\text{actual}}$ and the expected matrix $\mathbf{P}_{\text{sub}}$:

1. Calculate actual edge probabilities between communities $i$ and $j$:

$$\mathbf{A}_{\text{actual}}[i,j] = \begin{cases} \frac{\text{edge\_count}_{i,j}}{n_i(n_i-1)} & \text{if } i = j \\ \frac{\text{edge\_count}_{i,j}}{n_i \cdot n_j} & \text{if } i \neq j \end{cases} \tag{22}$$

where $n_i$ is the size of community $i$ and edge_count$_{i,j}$ is the number of edges between communities $i$ and $j$.

2. Compute mean absolute deviation: deviation $= \frac{1}{k^2} \sum_{i,j} |\mathbf{A}_{\text{actual}}[i,j] - \mathbf{P}_{\text{sub}}[i,j]|$

## I.2 SIGNAL STRENGTH METRICS

These metrics assess whether **within** each graph we can find meaningful, learnable community structure through different node-level predictive signals:

All signal metrics use Random Forest classification with the following configuration: 100 estimators, unlimited depth, minimum 2 samples per split, minimum 1 sample per leaf. Data is split 70/30 train/test with stratification to ensure all communities appear in both sets. Performance is measured using macro F1-score.

**Feature Signal**: Uses node features $\mathbf{x}_i$ directly as input to predict community labels.

**Degree Signal**: Uses node degree $d_i$ as single-dimensional input.

**Structure Signal**: For each node $v$, construct feature vector $\mathbf{f}_v \in \mathbb{R}^{3k}$ by concatenating community neighbor counts at distances 1, 2, and 3:

$$\mathbf{f}_v = [\mathbf{n}_v^{(1)}, \mathbf{n}_v^{(2)}, \mathbf{n}_v^{(3)}]$$

where $\mathbf{n}_v^{(d)} = [n_{v,1}^{(d)}, n_{v,2}^{(d)}, \ldots, n_{v,k}^{(d)}]$ and $n_{v,c}^{(d)}$ is the number of neighbors of node $v$ in community $c$ at exactly distance $d$. For example, with 5 communities, if node $v$ has neighbors in communities [1,1,2,4,4,4] at distance 1, [2,2,3,5,5,5,5] at distance 2, and [1,4,4] at distance 3, then $\mathbf{f}_v = [2, 1, 0, 3, 0, 0, 2, 1, 0, 4, 1, 0, 0, 2, 0]$.

### I.2.1 CROSS-GRAPH CONSISTENCY METRICS

These metrics evaluate whether community identities remain semantically consistent **across** different graph instances within a family:

**Structure Consistency**: For each graph $g$, compute

$$\text{consistency}_g = \frac{1}{k} \sum_{i=1}^{k} \rho_{\text{Spearman}}(\tilde{\mathbf{P}}_{i,:}, \mathbf{A}_{\text{actual},i,:}^{(g)})$$

where $\rho_{\text{Spearman}}$ denotes Spearman rank correlation, $\tilde{\mathbf{P}}_{i,:}$ is row $i$ of the universe propensity matrix restricted to participating communities, and $\mathbf{A}_{\text{actual},i,:}^{(g)}$ is the corresponding row of the actual probability matrix.

**Degree Consistency**: Combines within-graph and cross-graph consistency:

$$\text{consistency}_g = \frac{1}{2} \left( \rho_{\text{within}}^{(g)} + \rho_{\text{cross}}^{(g)} \right)$$

where $\rho_{\text{within}}^{(g)} = \rho_{\text{Spearman}}(\bar{\mathbf{d}}^{(g)}, \boldsymbol{\delta}_{\mathcal{C}^{(g)}})$ compares average degrees per community $\bar{\mathbf{d}}^{(g)}$ with universe degree centers $\boldsymbol{\delta}_{\mathcal{C}^{(g)}}$, and

$$\rho_{\text{cross}}^{(g)} = \frac{1}{\sum_{g' \neq g} w_{g,g'}} \sum_{g' \neq g} w_{g,g'} \cdot \rho_{\text{Spearman}}(\mathbf{s}^{(g)}, \mathbf{s}^{(g')})$$

where $\mathbf{s}^{(g)} \in [0,1]^K$ is the percentile signature for graph $g$ with $s_c^{(g)} = \frac{\text{rank}(\bar{d}_c^{(g)})-1}{|\mathcal{C}^{(g)}|-1}$ for participating communities and $s_c^{(g)} = \text{NaN}$ otherwise, $w_{g,g'} = |\{c : s_c^{(g)} \neq \text{NaN} \wedge s_c^{(g')} \neq \text{NaN}\}|$ is the overlap weight, and the correlation is computed only over non-NaN entries.

**Feature Consistency**: Average pairwise cosine similarity between community centroids:

$$\text{consistency} = \frac{2}{N(N-1)} \sum_{g=1}^{N-1} \sum_{g'=g+1}^{N} \frac{1}{k} \sum_{c=1}^{k} \frac{\boldsymbol{\mu}_c^{(g)} \cdot \boldsymbol{\mu}_c^{(g')}}{\|\boldsymbol{\mu}_c^{(g)}\| \|\boldsymbol{\mu}_c^{(g')}\|}$$

where $N$ is the number of graphs and $\boldsymbol{\mu}_c^{(g)}$ is the centroid of community $c$ in graph $g$.

| Category | Metric | Description |
|---|---|---|
| **Graph Property** | Homophily | Fraction of edges within communities: $h = \sum_{(i,j) \in E} \mathbb{1}[c(i) = c(j)]/|E|$ |
| | Average degree | Mean node degree, validating edge density scaling |
| | Degree tail ratio 99 | Ratio of 99th percentile to mean degree: $\tau_{99} = d_{99}/\bar{d}$ |
| | Generation time | Wall-clock time per graph instance |
| | Mean Probability Matrix Deviation | Average deviation between realized and target community edge probability matrices ($P_{\text{sub}}$) |
| **Signal Strength** | Feature signal | Per-graph, node-level community predictability via Random Forest (macro F1) using node features $\mathbf{x}_i$ as predictor |
| | Structure signal | Per-graph, node-level community predictability via Random Forest (macro F1) using $k$-hop neighbors label counts ($k \in \{1, 2, 3\}$) as predictor. |
| | Degree signal | Per-graph, node-level Community predictability via Random Forest (macro F1) using node degree $d_i$ as predictor. |
| **Cross-Graph Consistency** | Feature consistency | Average pairwise cosine similarity between community feature centroids across graphs |
| | Structure consistency | Spearman correlation between universe propensity matrix $\tilde{\mathbf{P}}$ and realized edge probabilities |
| | Degree consistency | Rank correlation of community degree orderings across graphs |

Table 3: Validation metrics for evaluating GraphUniverse generation framework.

## J    EXPANDED VALIDATION RESULT ANALYSIS

**Detailed Validation Result Analysis.** The correlation heatmap (Figure 2) demonstrates comprehensive parameter control across all validation metrics, revealing both expected relationships and theoretically interpretable effects.

**Graph Property Metrics** (first panel) show expected strong correlations with some additional insights. Input parameters precisely control observed homophily and average degree, while power-law exponent governs degree tail heaviness and node count correlates with generation time. We observe minor statistically significant effects on generation time from other parameters, likely reflecting computational complexity variations. Notably, increasing node count reduces probability matrix deviation, suggesting that larger graphs experience fewer random sampling effects due to improved statistical power. Slight deviations from target edge probability matrices under high average degree and degree separation parameters reflect the multiplicative edge generation process, where stronger degree factor effects naturally influence connection patterns.

**Signal Strength Metrics** (second panel) demonstrate strict adherence to theoretical expectations at the single-graph level, where Random Forest classifiers predict community labels within each graph instance. Cluster variance directly controls feature signal strength by determining feature separability between communities. The negative correlation between community count and all signal metrics reflects the fundamental difficulty of multi-class classification: distinguishing between two communities is inherently easier than discriminating among many, leading to higher F1 scores with fewer classes given similar discriminative power. Homophily's positive correlation with structure signal has a clear mechanistic explanation: when neighbors predominantly share the same community label, neighborhood composition becomes a highly predictive feature for node classification. This relationship—where averaging neighborhood representations provides strong community signals—likely explains the effectiveness of simple GNNs like GCN in homophilic settings. Similarly, average degree enhances structure signal by providing more neighborhood information, giving classifiers richer structural context for community prediction.

**Cross-Graph Consistency Metrics** (third panel) provide the strongest validation of our hierarchical design. The intended universe-level signals are present and tightly controllable: propensity variance governs structure consistency, degree separation controls degree consistency, and cluster

variance determines feature consistency. The negative correlation between community count and consistency metrics reflects increased sensitivity to random effects when computing correlations across many classes, where small sampling variations can more easily perturb rank orderings. We observe a theoretically justified trade-off between degree separation and propensity variance effects on edge probability deviation, emerging from our multiplicative edge generation process where $P_{ij} = \theta_i \theta_j P_{\text{scaled}}[c(i), c(j)]$, causing these parameters to modulate different components of the same generative mechanism.

## J.1 Individual Parameter-Validation Plots

Please see figures below.

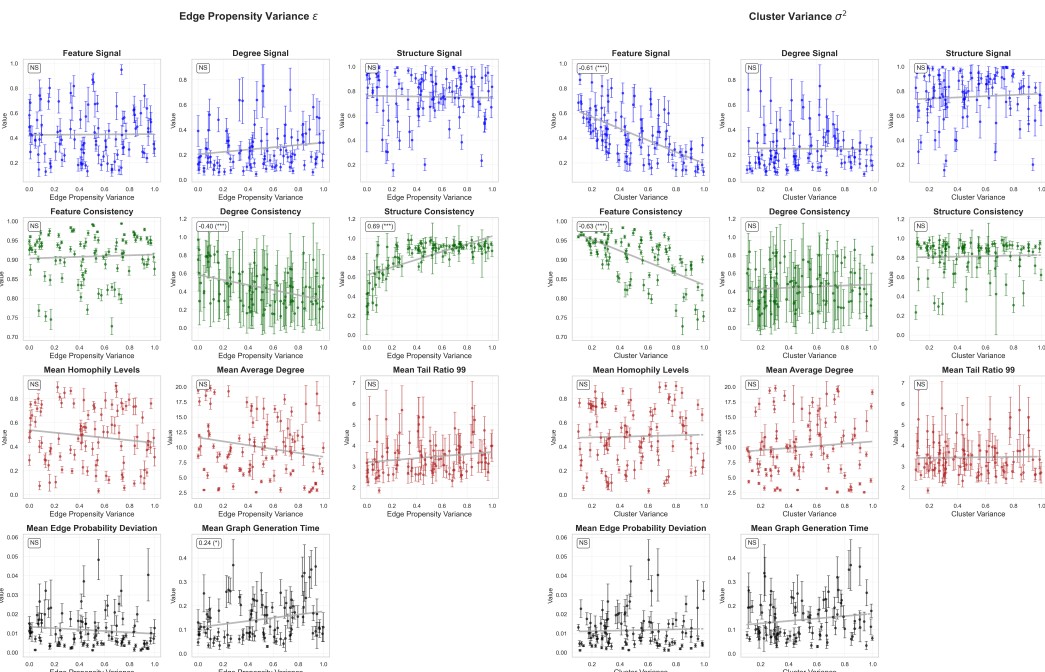

(a) Randomized Parameter Validation of Edge Propensity Variance parameter. Top left shows Pearson correlation and statistical significance level (NS, not statistically significant).

(b) Randomized Parameter Validation of Cluster Variance parameter. Top left shows Pearson correlation and statistical significance level (NS, not statistically significant).

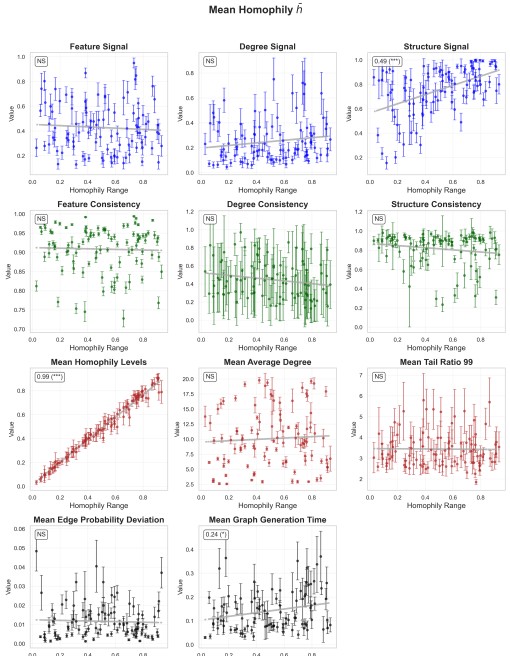

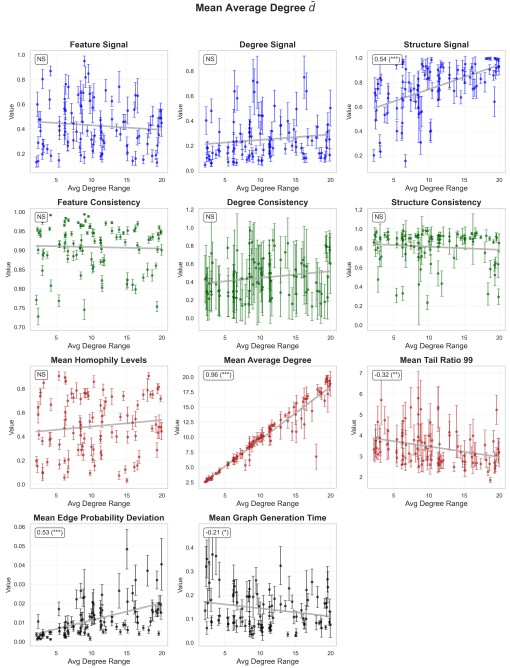

(a) Randomized Parameter Validation of Homophily Range parameter. Top left shows Pearson correlation and statistical significance level (NS, not statistically significant).

(b) Randomized Parameter Validation of Average Degree Range parameter. Top left shows Pearson correlation and statistical significance level (NS, not statistically significant).

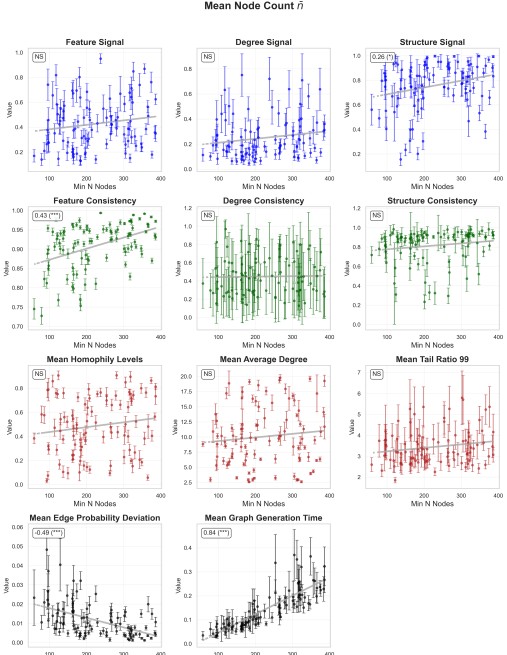

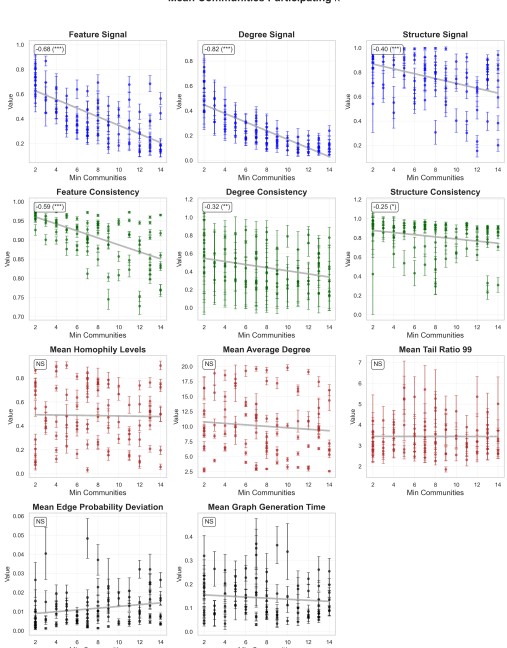

(a) Randomized Parameter Validation of Node Count range parameter. Top left shows Pearson correlation and statistical significance level (NS, not statistically significant).

(b) Randomized Parameter Validation of Communities Participating Per Graph Range parameter. Top left shows Pearson correlation and statistical significance level (NS, not statistically significant).

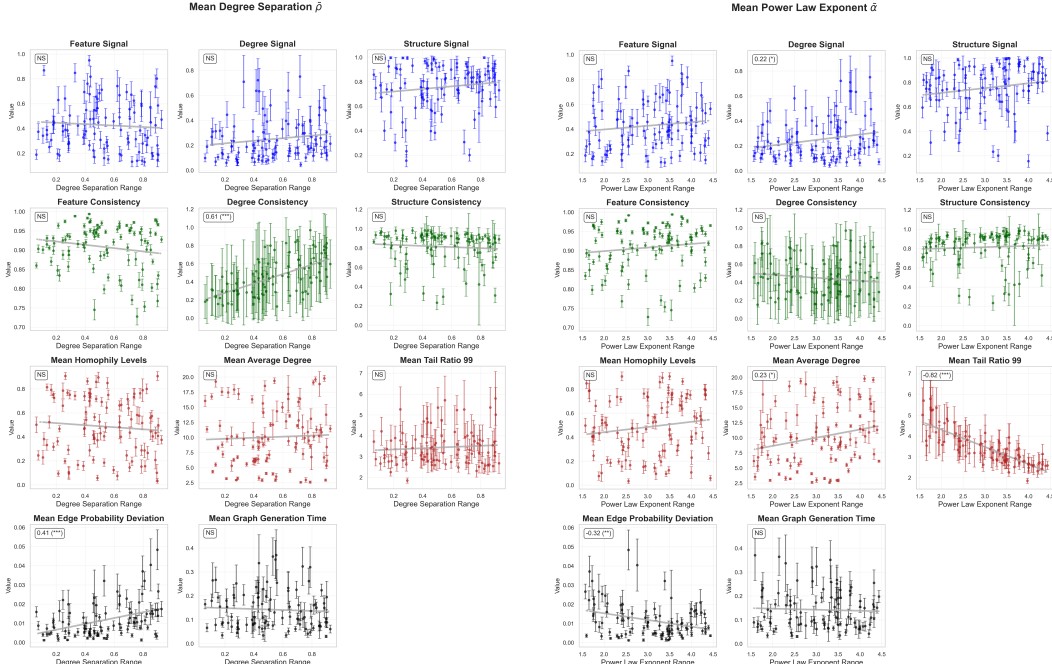

(a) Randomized Parameter Validation of Degree Separation Range parameter. Top left shows Pearson correlation and statistical significance level (NS, not statistically significant).

(b) Randomized Parameter Validation of Power Law Range parameter. Top left shows Pearson correlation and statistical significance level (NS, not statistically significant).

# K    REAL-WORLD DATASET VALIDATION

## K.1    EQUIVALENT DATASET PARAMETER EXTRACTION

To establish correspondence between real-world datasets and their synthetic equivalents, we developed a systematic parameter extraction pipeline that maps dataset characteristics to GraphUniverse and GraphWorld generation parameters.

### K.1.1    COMMUNITY DEFINITION STRATEGY

The extraction process begins by identifying a suitable notion of "community" for each dataset, which varies based on available features:

- **Molecular datasets (NCI1, MUTAG, AQSOL, OGBG-MolHIV):** We use atom types as natural communities, extracted via argmax on one-hot encoded features or directly from atomic number features.
- **IMDB-MULTI:** Lacking node features or labels, we perform degree-based clustering using K-means on node degree and clustering coefficient features, with optimal $K$ determined via silhouette scoring.

### K.1.2    PARAMETER MAPPING METHODOLOGY

For each dataset, we extract:

1. **Graph size distribution:** Number of nodes across all graphs
2. **Average degree distribution:** Mean degree per graph
3. **Community structure:** Number of unique communities per graph and total unique communities
4. **Homophily:** Fraction of edges connecting nodes in the same community

For **GraphUniverse**, we use the 5th-95th percentile range of each property to capture the full distribution while avoiding outliers. The universe size $K$ is set to the maximum of (i) the 90th percentile of communities needed to cover 90% of nodes, and (ii) the maximum communities per graph, ensuring sufficient diversity.

For **GraphWorld**, following their single-graph paradigm, we use mean values for all properties. Both $K$ and communities per graph are set to the dataset's mean unique communities per graph. However, for fairness, we set the graph size to 1000 instead of the average of the real dataset's graph sizes, which in general are too small in an inductive dataset to train a model on.

### K.1.3 EXTRACTED PARAMETERS

Table 4 presents the extracted parameters for all datasets. Note that GraphUniverse captures the heterogeneity of real datasets through ranges, while GraphWorld reduces this to point estimates.

Table 4: Extracted parameters from real datasets and their synthetic equivalents. GraphUniverse uses 5th-95th percentile ranges; GraphWorld uses mean values.

| Dataset | #Graphs | Nodes GU Range | GW | Avg. Degree GU Range | GW | Homophily GU Range | GW | Communities/Graph GU Range | GW | Universe K |
|---|---|---|---|---|---|---|---|---|---|---|
| OGBG-MolHIV | 41,127 | [13, 46] | 1000 | [2.00, 2.50] | 2.14 | [0.32, 0.84] | 0.61 | [2, 5] | 3 | 5/3 |
| AQSOL | 9,833 | [10, 36] | 1000 | [1.60, 2.25] | 1.98 | [0.15, 0.92] | 0.59 | [2, 5] | 2 | 5/2 |
| IMDB-MULTI | 1,500 | [10, 31] | 1000 | [4.67, 17.00] | 8.10 | [0.35, 1.00] | 0.80 | [2, 5] | 2 | 5/2 |
| NCI1 | 4,110 | [15, 59] | 1000 | [2.00, 2.50] | 2.16 | [0.38, 0.82] | 0.62 | [2, 5] | 3 | 5/3 |
| ZINC | 10,000 | [16, 31] | 1000 | [2.00, 2.50] | 2.14 | [0.32, 0.71] | 0.52 | [3, 6] | 4 | 6/4 |

**Notes:** GU = GraphUniverse, GW = GraphWorld. For Universe K, we show both values (GraphUniverse/GraphWorld) when they differ. For all experiments except the IMDB-MULTI one, other generation parameters (edge propensity variance, cluster variance, degree separation) are set to mid-range values (0.5) for consistency across experiments. For the IMDB-MULTI one, since it does not have any node-features, we set the feature signal to zero in equivalent dataset (center variance of 0.01, cluster variance of 1.0) and the degree separation, now being the main identifier for community detection, to a range of 0.9 to 1.0.

## K.2 DETAILED REAL-WORLD ALIGNMENT ANALYSIS

Figure 10 provides a detailed model-level analysis of ranking correlations between synthetic and real-world datasets. For each of the six datasets, we plot individual model rankings computed via bootstrap analysis (1000 iterations) to quantify uncertainty in rank estimates. In this plot we omitted the baseline models, since these artificially inflate correlation. This effect is displayed in Figure 11, where the DeepSet and GraphMLP models *are* included.

The results confirm that GraphUniverse preserves real-world model ranking patterns across diverse datasets. GraphUniverse (top row) achieves strong positive Spearman correlations, demonstrating that models maintaining similar relative performance in both synthetic and real settings. In contrast, GraphWorld (bottom row) shows poor alignment with multiple negative correlations, failing to capture how model performance varies across different graph structures. This ranking preservation is crucial for practitioners who need to select architectures based on synthetic benchmark results, as GraphUniverse reliably predicts which models will perform well on real-world tasks.

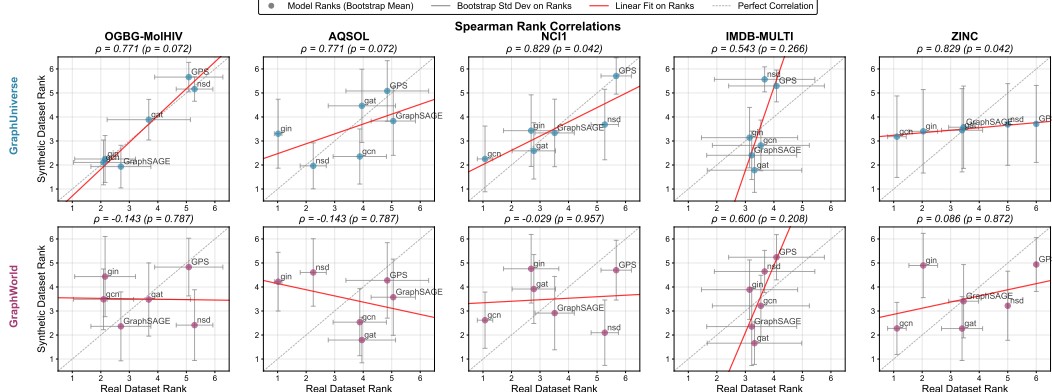

Figure 10: Model ranking correlations between real and synthetic datasets. Each point represents a model's rank (1=best) with error bars showing bootstrap standard deviation (1000 iterations). GraphUniverse (top row) demonstrates strong rank preservation, while GraphWorld (bottom row) shows poor alignment including negative correlations. The red line shows linear fit to mean ranks, with the diagonal gray line indicating perfect correlation. Non-message passing models (DeepSet and GraphMLP) are omitted to focus on graph-aware architectures, as they consistently underperform and artificially inflate correlations.

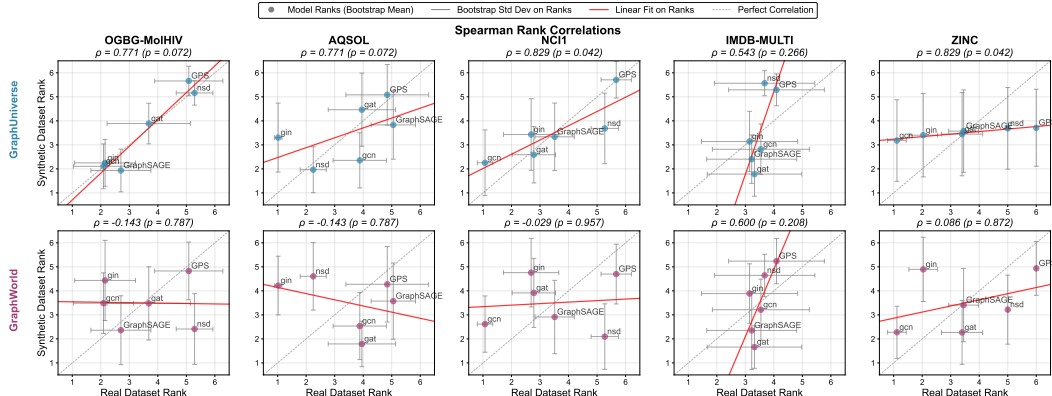

Figure 11: Model ranking correlations between real and synthetic datasets. Each point represents a model's rank (1=best) with error bars showing bootstrap standard deviation (1000 iterations). The red line shows linear fit to mean ranks, with the diagonal gray line indicating perfect correlation. Non-message passing models (DeepSet and GraphMLP) are inceled, artificially inflating correlations, especially for the GraphWorld case (bottom row).

## L HETEROPHILY-SPECIALIZED ARCHITECTURES EVALUATION

To further validate our framework's ability to capture nuanced architectural differences across graph properties, we extend our benchmarking to include models that have shown effectiveness in heterophilic settings: Frequency Adaptive Graph Convolutional Network (FAGCN) (Bo et al., 2021), H2GCN (Zhu et al., 2020), and ChebNet (Tang et al., 2024). While FAGCN and H2GCN were explicitly designed for heterophilic graphs, ChebNet's spectral approach using Chebyshev polynomials has demonstrated strong empirical performance in low-homophily settings, making it a valuable addition to our analysis.

### L.1 EXPERIMENTAL SETUP

We evaluate FAGCN, H2GCN, and ChebNet alongside our original model suite across five homophily levels (0.05, 0.25, 0.5, 0.75, 0.95) in the transductive setting (*GraphWorld* style) and three homophily

ranges ([0.0,0.1], [0.4,0.6], [0.9,1.0]) in the inductive setting (*GraphUniverse* style). All other experimental parameters remain consistent with our main benchmarking protocol (Section 5.1), including hyperparameter optimization procedures and evaluation metrics.

The hyperparameter grid used in optimization of these heterophily-specialized models is displayed in Table 5.

## L.2 RESULTS AND ANALYSIS

Figure 12 presents the performance of heterophily-specialized models compared to our baseline architectures across the homophily spectrum.

Our results reveal surprising differences between evaluation paradigms:

**Transductive Performance:** As expected, H2GCN and ChebNet demonstrate superior performance in the most heterophilic regimes (0.05-0.25), validating their design principles for heterophilic graphs. Their advantage gradually diminishes as homophily increases, with performance converging to baseline levels at high homophily (0.95). Notably, FAGCN underperforms relative to expectations, suggesting that its frequency-adaptive mechanisms may not translate effectively to our synthetic community detection task.

**Inductive Performance:** In contrast, none of the heterophily-specialized models maintain their advantages in the inductive setting. Performance differences across homophily levels become less pronounced, and specialized architectures show no clear superiority over standard message-passing networks like GAT and GraphSAGE. This finding suggests that the benefits of heterophily-specific designs may be diminished when models must generalize to entirely new graph instances rather than leveraging global structural patterns within a single graph.

This experiment further validates GraphUniverse's capacity to reveal architectural behaviors that remain hidden in traditional single-graph evaluations. Architectural advantages observed in transductive settings may not transfer to inductive scenarios, emphasizing the importance of evaluation paradigm choice.

Table 5: Hyperparameter grid search space for heterophily-specialized models.

| Model | Hyperparameter | Search Space (Grid) |
|---|---|---|
| ChebNet | feature_encoder.out_channels | {32, 64} |
| | feature_encoder.proj_dropout | {0.3} |
| | backbone.num_layers | {2, 4} |
| | backbone.K | {2, 3, 5} |
| | backbone.normalization | {sym, rw} |
| | backbone.dropout | {0.2, 0.4} |
| | readout.hidden_layers | {[16], []} |
| | readout.dropout | {0.3} |
| FAGCN | feature_encoder.out_channels | {32, 64} |
| | feature_encoder.proj_dropout | {0.3} |
| | backbone.num_layers | {2, 4} |
| | backbone.eps | {0.0, 0.1, 0.2} |
| | backbone.normalize | {True, False} |
| | backbone.dropout | {0.2, 0.4} |
| | readout.hidden_layers | {[16], []} |
| | readout.dropout | {0.3} |
| H2GCN | feature_encoder.out_channels | {32, 64} |
| | feature_encoder.proj_dropout | {0.3} |
| | backbone.num_layers | {2, 4} |
| | backbone.k | {2, 3} |
| | backbone.use_relu | {True, False} |
| | backbone.dropout | {0.2, 0.4} |
| | readout.hidden_layers | {[16], []} |
| | readout.dropout | {0.3} |

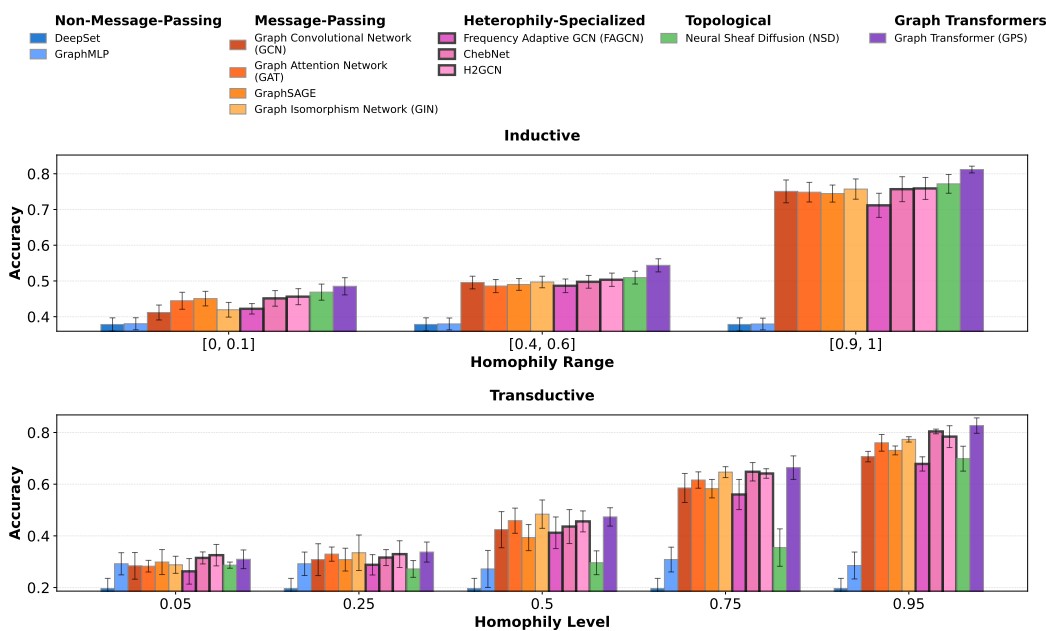

Figure 12: Performance comparison of heterophily-specialized models (FAGCN, H2GCN, ChebNet) against baseline architectures across varying homophily levels in inductive and transductive settings. Error bars represent the standard deviation of the test performance across different random seeds.

## M    FURTHER EXPERIMENTAL DETAILS

This appendix provides supplementary details regarding the experimental setup used in this work to ensure reproducibility.

### M.1    GRAPH GENERATION PARAMETERS

Table 6 specifies the default generation parameters used for both the primary inductive setting and the baseline transductive setting. The universe parameters define the underlying semantic space (e.g., number of communities, feature characteristics), while the family parameters control the structural properties of the sampled graphs. Note the key differences: the inductive setting generates a large family of smaller, varied graphs, whereas the transductive setting generates a single, large graph with fixed properties.

### M.2    BENCHMARKED MODEL ARCHITECTURES

This section provides a brief overview of the models included in our experimental evaluation.

**GraphMLP & DeepSet**    These models serve as non-message-passing baselines. **DeepSet** is a permutation-invariant architecture for learning on sets, ignoring all structural information (Zaheer et al., 2017). **GraphMLP** basically extends DeepSet by incorporating graph structure during training via a neighborhood contrastive loss, which encourages linked nodes to have similar representations (Hu et al., 2021).

**Graph Convolutional Network (GCN)**    The GCN is a foundational GNN architecture that learns node representations by efficiently aggregating feature information from its immediate neighbors through a spectral-based graph convolution (Kipf & Welling, 2017).

**GraphSAGE**    Short for Graph SAmple and aggreGatE, this model provides a framework for inductive node embedding (Hamilton et al., 2017). Instead of training on the entire graph, it learns

aggregation functions on a fixed-size sample of a node's neighborhood, allowing it to generalize to unseen nodes and graphs.

**Graph Isomorphism Network (GIN)**  The GIN is a powerful GNN designed to be as discriminative as the Weisfeiler-Lehman (WL) graph isomorphism test (Xu et al., 2019). It achieves this by using an MLP to update node features, making it highly effective for tasks requiring a strong understanding of graph structure.

**Graph Attention Network v2 (GATv2)**  GATv2 (Brody et al., 2022) is an improved version of the original GAT model (Veličković et al., 2018) that uses a modified attention mechanism to make it strictly more expressive. By assigning different importance weights to different nodes in a neighborhood, both GAT and GATv2 can focus on the most relevant parts of the graph for a given task,

**Neural Sheaf Diffusion (NSD)**  NSD generalizes message passing to cellular sheaves, which are topological structures capable of representing more complex relationships (Bodnar et al., 2022). This allows it to capture richer structural and relational information than standard GNNs.

**TopoTune**  TopoTune is a framework designed to systematically generalize any GNN into a topo-logical neural network, making higher-order structures accessible for learning (Papillon et al., 2025). It operates by taking a GNN as input and using it as a building block within a more expressive architecture called a Generalized Combinatorial Complex Network (GCCN). This is achieved by expanding a higher-order structure (like a simplicial or cell complex) into a collection of graphs, which are then processed by an ensemble of synchronized GNNs. This approach democratizes topological deep learning by allowing practitioners to easily "upgrade" existing GNNs to reason about complex, multi-way relationships beyond simple edges.

**GPS (Graph Transformer)**  The GPS model combines the expressive power of transformers with standard message-passing GNNs (Rampášek et al., 2022). By integrating local structural information with global attention mechanisms and positional encodings, it aims to capture a wide range of dependencies in the graph, making it a very powerful and flexible architecture.

### M.3    Hyperparameter Optimization

For each model, we leverage the TopoBench infrastructure (Telyatnikov et al., 2025) to perform an extensive grid search to identify optimal hyperparameter configurations, optimizing for the highest mean accuracy on a held-out validation set (over three dataset seeds). Table 7 details the complete search space used for every hyperparameter of each model (following TopoBench logic of feature encoder, backbone and readout modules), providing a basis for the reproducibility of our experiments. Unless otherwise specified in Table 7, we note that sum is the by default pooling method.

**Remark.** It should be noted that these spaces were refined based on preliminary, larger-scale grid searches; we pruned parameter options that consistently showed a negligible or detrimental impact on performance to focus on the most influential hyperparameters. Furthermore, to limit the combinatorial explosion of the search space, parameters such as batch size and optimizer settings were fixed. These values were informed by previous benchmarks in TopoBench and TopoTune (Papillon et al., 2025); for example, we used the Adam optimizer with a learning rate of 0.001 and set the batch size to 32 for all inductive experiments.

### M.4    Hardware Details

The hyperparameter search is executed on a Linux machine with 256 cores, 1TB of system memory, and 4 NVIDIA H100 GPUs, each with 94GB of GPU memory.

Table 6: Default **GraphUniverse** generation parameters for Inductive and Transductive settings. Differences are highlighted in bold.

| | Inductive Value | Transductive Value |
|---|---|---|
| **Universe Parameters** | | |
| Number of communities ($K$) | 10 | 10 |
| Feature Dimension | 15 | 15 |
| Center Variance ($\sigma_{\text{center}}^2$) | 0.2 | 0.2 |
| Cluster Variance ($\sigma_{\text{cluster}}^2$) | 0.5 | 0.5 |
| Edge Propensity Variance ($\epsilon$) | 0.5 | 0.5 |
| Seed | 42 | 42 |
| **Family Parameters** | | |
| Number of graphs | **1000** | **1** |
| Min Node Count ($n_{\min}$) | **50** | **1000** |
| Max Node Count ($n_{\max}$) | **200** | **1000** |
| Min Communities ($k_{\min}$) | **4** | **10** |
| Max Communities ($k_{\max}$) | **6** | **10** |
| Homophily Range ($h_{\min}, h_{\max}$) | **[0.4, 0.6]** | **[0.5, 0.5]** |
| Average Degree Range ($d_{\min}, d_{\max}$) | **[1.0, 5.0]** | **[2.5, 2.5]** |
| Degree Separation Range ($\rho_{\min}, \rho_{\max}$) | **[0.5, 0.8]** | **[0.5, 0.5]** |
| Degree distribution | `power_law` | `power_law` |
| Power Law Exponent Range ($\alpha_{\min}, \alpha_{\max}$) | **[2.0, 2.5]** | **[2.5, 2.5]** |
| Seed | (Inherited from Universe) | |

Table 7: Hyperparameter grid search space for each model.

| Model | Hyperparameter | Search Space (Grid) |
|---|---|---|
| **GCN** | `feature_encoder.out_channels` | {32, 64} |
| | `backbone.num_layers` | {2, 4} |
| | `backbone.dropout` | {0.2, 0.4} |
| | `readout.hidden_layers` | {[16], []} |
| **GIN** | `feature_encoder.out_channels` | {32, 64} |
| | `backbone.num_layers` | {2, 4} |
| | `backbone.dropout` | {0.2, 0.4} |
| | `readout.hidden_layers` | {[16], []} |
| **GraphSAGE** | `feature_encoder.out_channels` | {32, 64} |
| | `backbone.num_layers` | {2, 4} |
| | `backbone.dropout` | {0.2, 0.4} |
| | `readout.hidden_layers` | {[16], []} |
| **GAT** | `feature_encoder.out_channels` | {32, 64} |
| | `backbone.num_layers` | {2, 4} |
| | `backbone.heads` | {2, 4, 8} |
| | `backbone.dropout` | {0.0, 0.2} |
| | `readout.hidden_layers` | {[16], []} |
| **GPS** | `feature_encoder.out_channels` | {32, 64} |
| | `backbone.num_layers` | {2, 4} |
| | `backbone.heads` | {4} |
| | `backbone.dropout` | {0.2, 0.4} |
| | `backbone.attn_type` | {`multihead`, `performer`} |
| | `transforms.encodings` | {RWSE, LapPE} |
| | `readout.hidden_layers` | {[16], []} |
| **NSD** | `feature_encoder.out_channels` | {32, 64} |
| | `backbone.num_layers` | {2, 4, 6} |
| | `backbone.dropout` | {0.2, 0.4} |
| | `backbone.sheaf_type` | {`bundle`, `diag`} |
| | `transforms.encodings` | {RWSE, LapPE} |
| | `readout.hidden_layers` | {[16], []} |
| **GraphMLP** | `feature_encoder.out_channels` | {32, 64} |
| | `backbone.order` | {2, 4} |
| | `backbone.dropout` | {0.2, 0.4} |
| | `readout.hidden_layers` | {[16], []} |
| **DeepSet** | `feature_encoder.out_channels` | {32, 64} |
| | `readout.hidden_layers` | {[64, 32], [32, 16], [16]} |
| | `readout.dropout` | {0.2, 0.4} |
| **TopoTune** | `model_type` | {`cell`, `simplicial`} |
| | `feature_encoder.out_channels` | {32, 64} |
| | `tune_gnn` | {GCN, GIN, GAT, GraphSAGE} |
| | `backbone.layers` | {2, 4} |
| | `readout.pooling_type` | {mean, sum} |
| | `backbone.neighborhoods` | {10 predefined topological operator sets} |

