# OpenReview forum: "GraphUniverse: Synthetic Graph Generation for Evaluating Inductive Generalization"
_ICLR.cc/2026/Conference — ICLR 2026 Poster_

### Official Review · Reviewer_3ZL3 · 2025-10-18

**Soundness:** 2
**Presentation:** 3
**Contribution:** 2
**Rating:** 4
**Confidence:** 5

**Summary:**

GraphUniverse is a hierarchical generator for families of graphs designed to study inductive generalization across unseen graphs. At the “universe” level it fixes persistent communities with heterogeneous edge propensities, degree profiles, and feature centroids; “family” ranges then constrain homophily, average degree, size, and degree separation; graph instances sample from these ranges and realize edges via a Bernoulli reformulation of DC‑SBM with community‑coupled degree factors and feature Gaussians. Validation correlates generation parameters with graph properties, signal strength, and cross‑graph consistency. Benchmarking compares inductive vs. transductive performance and robustness under controlled distribution shifts and size scaling for several architectures.

**Strengths:**

1. A very timely and important direction for the whole graph community
2. the demo website is well designed and validates the reproducibility
3. comprehensive experiments

**Weaknesses:**

1. the biggest problem of this framework is that I don't see its clear advantage over other generative frameworks like GraphWorld, CGT (Graph Generative Model for Benchmarking Graph Neural Networks, icml 2023). And it doesn't mention a very relevant work, A Metadata-Driven Approach to Understand Graph Neural Networks (nips 2023). Although authors conduct comprehensive experiments on using this framework to evaluate different kinds of GML architectures, two experiments are missing: (1) the horizontal comparison between this framework against other generative pipelines, this makes the value of this work unclear; (2) This paper presents train on syn, test on real-world experiments, but there's no train on syn, test on real-world ones. This limits the usage of these synthetic datasets. I also doubt whether these purely synthetic graphs can really be used to pressure test GNN models. I personally think the way CGT is adopting (start from real-world ones and do generation) is more promising.
2. In the very beginning, authors talk about the problem of graph foundation model. However, i don't see this work able to address these relevant problems.
3. Statements about enabling “next‑generation graph foundation models” are aspirational without pre‑training results (Abstract; Sec. 1, 6). No direct evidence found in the manuscript.
4. Semantic “consistency” is validated via statistical proxies, not via downstream cross‑family transfer beyond the reported tasks (Fig. 2; Sec. 4.2–5).
5. Why are these factors considered? There's no validation. You can see (Metadata-Driven Approach to Understand Graph Neural Networks (nips 2023)) conducts a statistical test, which is much more promising and scientific.
6. Table 1 reports only two sizes (100 vs. 1000 nodes) and single‑thread CPU throughput; linearity claims rest on a very small sample (Sec. 4.3; Table 1)—limited generality.
7. Minor notation/typo inconsistencies (e.g., “DCCC‑SBM true true” in Table 4)
8. Only one transformer variant (GPS) is included (Sec. 5.1), narrowing architectural coverage. GraphGPS also heavily relies on GNN.

**Questions:**

Please see weakness

---

> ### Author Response · Authors · 2025-11-20
> **Response to Reviewer 3ZL3 (1/2)**
>
> We sincerely thank the reviewer for their insightful and constructive feedback. We appreciate the positive comments on the potential impact on the community of our framework for inductive evaluation and the utility of the interactive demo. The concerns raised are helpful and valid, opening up interesting discussions. We aim to address each below, organized per topic:
>
> **On Framework Advantages and Related Work.** The reviewer brings up two insightful and relevant works. Therefore, we have now included CGT and the metadata-driven approach paper in our related work section as they represent valuable contributions to synthetic graph generation and study of model behaviour in function of structural graph properties. Nevertheless, there is one major concept that differs our approach from these papers, being that both are confined to transductive, single-graph settings (in the real-world graphs analyzed and generated). GraphUniverse's primary contribution is extending this paradigm to inductive evaluation, enabling systematic studies that are fundamentally impossible with existing frameworks: controlled distribution shift analysis (RQ2) and cross-graph size generalization (RQ3).
>
> **On Real-World Validation.** Another strength of the papers mentioned by the reviewer, is their direct relation with real-world datasets. We agree with the concerns about the gap between synthetic and real-world performance. As a result, we conducted an additional experiment (RQ4, Subsection 5.5 and Appendix K) that we hope addresses this limitation. In this experiment, we selected five real-world inductive graph-level datasets (OGBG-MolHIV, AOQSOL, NCI1, IMDB-MULTI, ZINC) and generated corresponding GraphUniverse families and GraphWorld by matching the distributions of their key structural properties (range of properties for GraphUniverse and mean property for single-graph GraphWorld version). We then trained our full suite of models and computed the correlation between the model rankings on the real datasets and their GraphUniverse and GraphWorld equivalents (full details in Appendix K). We were happy to find that the results show a strong and consistent positive correlation with the GraphUniverse equivalent dataset, while the GraphWorld equivalent datasets show low to negative correlation on most datasets, demonstrating that model performance and ranking trends observed in GraphUniverse are indeed predictive of inductive performance in real-world settings, in contrast to the single-graph approach.
>
> **Clarifying the Positioning Regarding Graph Foundation Models.** The reviewer raises a valid point about our foundation model claims being premature. We acknowledge that without direct pre-training experiments, some of our wording is indeed aspirational. In response to this, we have revised the abstract, introduction, and conclusion to focus on our core contribution: providing systematic inductive evaluation capabilities that have been missing from the field. However, we note that recent influential works [1,2] explicitly call for exactly these types of inductive evaluation frameworks for GFM development, suggesting there is a clear potential avenue for our work to contribute. In response to this alignment and another reviewer's request for substantial discussion about GFM applications, we have added a detailed discussion in appendix B, so not in the main text, outlining how our framework could contribute in this field. We hope this approach—moderating our claims while still discussing potential future directions—appropriately balances the concerns raised.
>
> **Minor Fixes.** We have corrected notation and typo inconsistencies throughout the manuscript and expanded our scalability analysis (Table 1) with additional data points to better support our linearity claims.
>
> **On Semantic Consistency Validation.** We acknowledge that our validation relies on statistical proxies rather than downstream cross-family transfer. We chose this approach as an internal validation that universe-level information is preserved across generated graphs and responds predictably to parameter changes. We hope our new RQ4 experiment provides convincing external validation of our framework's utility.
>
> [1] Bechler-Speicher, Maya, et al. "Position: Graph Learning Will Lose Relevance Due To Poor Benchmarks." Forty-second International Conference on Machine Learning Position Paper Track.
>
> [2] Wang, Zehong, et al. "Graph Foundation Models: A Comprehensive Survey." arXiv preprint arXiv:2505.15116 (2025).
>
> (1/2)

---

> > ### Author Response · Authors · 2025-11-20
> > **Response to Reviewer 3ZL3 (2/2)**
> >
> > **On Factor Selection and Statistical Validation.** The reviewer makes a valid point about our directed approach to factor selection versus comprehensive statistical testing. We chose a targeted, RQ-based methodology instead, focusing on underexplored questions like distribution shift robustness, while needing less computational resources. We acknowledge this represents a limitation and hope that with greater computational resources we could have the framework deployed for fully randomized studies with explanatory linear models, as suggested.
> >
> > **On Limited Transformer Coverage.** We chose GPS because it encompasses multiple attention mechanisms and represents a leading transformer approach for graph learning–in fact, it is consistently the best performing model in our experimental section, already showing the power on transformer architectures. That being said, we acknowledge the value of broader architectural diversity, and are to incorporate FAGCN [3] into our analysis before the rebuttal ends (a heterophily-specialized model, suggested by reviewer Wz29). However, we note that comprehensive model coverage was not a primary objective of this work, as our focus centers on demonstrating the systematic evaluation capabilities enabled by our GraphUniverse.
> >
> > Once again we thank the reviewer for the insightful and constructive criticism. We hope that the changes we made addressed your concerns and would be happy to discuss any of these points further.
> >
> > [3] Bo, Deyu, Xiao Wang, Chuan Shi, and Huawei Shen. "Beyond low-frequency information in graph convolutional networks." In Proceedings of the AAAI conference on artificial intelligence, vol. 35, no. 5, pp. 3950-3957. 2021.
> >
> > (2/2)

---

> > > ### Author Response · Authors · 2025-11-26
> > >
> > > Following up on our previous response regarding the inclusion of heterophily-specialized models (suggested by Reviewer **Wz29**), we are pleased to share the initial results for FAGCN across varying homophily levels in both inductive and transductive settings:
> > >
> > > https://postimg.cc/2LRK6G32
> > >
> > > As shown, FAGCN performance does not stand out in heterophilic regimes; it outperforms GCN in the inductive setting, but surprisingly it does not perform well in the transductive one. Given the observed results, we are currently extending the experiments for other heterophilic models and will shortly upload a revised manuscript incorporating these full results shortly.
> > >
> > > Once again, we appreciate the reviewer's thoughtful feedback and welcome any additional questions or comments.

---

> > > > ### Comment · Reviewer_3ZL3 · 2025-11-27
> > > >
> > > > Thanks for the response. I have changed my scores accordingly.

---

> > > > > ### Author Response · Authors · 2025-11-28
> > > > >
> > > > > We thank the reviewer for his thoughtful engagement and for taking the time to reconsider our revisions. We appreciate your constructive feedback, which helped us strengthen the paper.

---

### Official Review · Reviewer_N9mv · 2025-10-23

**Soundness:** 3
**Presentation:** 2
**Contribution:** 3
**Rating:** 6
**Confidence:** 4

**Summary:**

This paper presents a framework to generate graphs to evaluate graph learning models on transductive and inductive tasks. Furthermore, the authors use their tool to benchmark a variety of GNNs and achieve results about the impact of data on model generalization (e.g. the impact of distribution shifts).

**Strengths:**

- **(S1) Originality + Significance:** Current benchmarks of GNNs are severely lacking (as pointed out by _[Position: Graph learning will lose relevance due to poor benchmarks](https://arxiv.org/abs/2502.14546)_). GraphUniverse comes timely to help with this issue. It seems to be easy to use (please contrast with W3) and can be used to create a diverse array of graphs. This could lead the community to a more principled evaluation of GNN architectures and thus to a better understanding of their capabilities.
- **(S2) Clarity:** The paper is well written and easy to understand. (Please contrast with W2)
- **(S3) Quality:** The authors use GraphUniverse to benchmark a variety of different GNNs. I am impressed by the variety of benchmarked GNNs and find the results interesting:
	- GIN struggles in the transductive setting (but not the inductive) whereas the graph transformer performs similarly in both settings. This is interesting to me, as I would have expected GIN to also perform well in the inductive case. Furthermore, while the explanation _... indicating its success may stem from memorizing a single graph’s structure_ makes intuitive sense, it goes against my experience of GIN performing well on inductive molecular tasks.
	- Furthermore, the benchmarks show that models are less robust to distribution shifts than thought. This opens the research direction of developing models that perform better under distribution shifts.

**Weaknesses:**

I like the contributions of this work and my weaknesses all center around the presentation.

- **(W1) Citations:** This paper often cites the arXiv version of papers instead of the published version. While not only formally wrong, this makes it difficult to evaluate whether this paper builds upon established research or on irrelevant work.
- **(W2) Presentation of experimental results:** Figure 3 is the cornerstone of the experimental section. It contains almost all results and most research questions reference it. Despite this, Figure 3 is extremely unclear making it difficult to verify the claims of the authors. Below are some pointers for improvement that could serve as inspiration to make the figure more legible:
	- The first line reads "Transductive" and the second "Inductive". However, in the legend you have a separate style for inductive and transductive results and according to this the first result in the transductive line is inductive.
	- The split into (A) and (B) is not that clear. It would help if you could explicitly mention in the figure (not in the caption!) which research question (A) and (B) answer.
	- Explicit color coding and re-use: When your main text refers to the result of a model in Figure 3 re-use the color of the model in the main text. This makes it easier to find that information in the figure. Furthermore, you could explicitly color code types of models (which I think you attempted but it is not clear enough).
	- Why do some results have N/A?
	- Alternatively, you could simplify this figure by showing the results of fewer models (and putting the more extensive figure in the appendix). As far as I can see, you are not discussing the results of GCN, GAT and TopoTune in the main text anyway.
- **(W3) No code:** While the authors pledge to make the code public there is no code for the reviewers to check. It is unclear how easy to use GraphUniverse is. Furthermore, I wanted to check the code for the following. You write:
>Graph-level tasks (triangle counting) initially show only GPS, NSD, and GIN effectively solving the task, but while GPS and NSD maintain performance when scaling to larger graphs, GIN fails to generalize, suggesting traditional MPNNs overfit to training graph sizes.


- **(W4) Figures of graphs:** Fundamentally, this paper is about generating graph data. Thus, the  paper should include some figures of generated graphs. While the interactive demo is great, it is not guaranteed to be online or work in the future.

Is it possible that these models differ in how they aggregate local embeddings to the global level (e.g. sum vs mean pooling)?

**Questions:**

- See weaknesses


**Overall,** I think this is a good paper with clear flaws in its presentation. Thus, I vote for (6 - marginally above acceptance threshold).

### Miscellaneous
- Section 4.1.1 should in my opinion be a subsection instead of the only sub-subsection in the main paper.
- No information about the pooling type for some architectures.

---

> ### Author Response · Authors · 2025-11-20
> **Response to Reviewer N9mv**
>
> We sincerely thank you for your thorough and constructive review. We appreciate your support for our core contributions and are grateful for your detailed feedback on presentation issues that are essential for meeting ICLR's professional standards. We have addressed your concerns through the following revisions:
>
> **Regarding Citations (W1).** Thank you for identifying this issue. We have updated arXiv citations to their published versions where available.
>
> **Regarding the Presentation of Experimental Results (W2).** We agree this figure requires more clarity. Each of your suggestions substantially improved readability, thus we are grateful for these specific recommendations. We have implemented the following changes:
> * We fixed the hash inconsistency of the first subplot of the first row (transductive)
> * We moved the (A) and (B) split on the left to the already existing subtitles.
> * We chose better colors for model family-coding and used the new colors for text coloring when referring to models in the main text. Moreover, we adapted the legend of the main figure to now be organized per model family (subtitles within the caption).
> * We specified in the caption that N/A is used in settings where the distribution shift would lead to input parameters outside parameter bounds (like for example homophily under zero).
>
> **Regarding the Code Release (W3).** As specified in our general comment, we release the full GraphUniverse graph generation code under an anonymous link (https://anonymous.4open.science/r/GraphUniverse-3458). Upon acceptance, we plan to release this as a PyPi package. This and more is detailed in the new Reproducibility Section added right before the reference list.
>
> **Regarding Graph Visuals (W4).** We agree that the paper would look more visually appealing when actually seeing the graphs. However, we are currently constrained by space limitations and have allocated the additional rebuttal page to our new validation experiment. We will assess whether sufficient space can be made available once the final text is finalized near the end of the rebuttal phase. In the meantime, sample visualizations are available in our anonymous GitHub repository. Regarding the demo's long-term maintenance, we have mitigated this risk by providing all necessary code to run the Streamlit interface locally with a single command. This is available both in the anonymous GitHub repository (streamlit_graph_universe.py) and will be included in the PyPI package upon acceptance, ensuring researchers can access the interface regardless of the online demo's status.
>
> **Miscellaneous.** Thank you for the suggestions, we have addressed both points in the revised manuscript: previous Sub-subsection 4.1.1 is now Subsection 4.1, and we specify in Appendix L.3 that sum is the default pooling operation used in all architectures (unless otherwise specified in Table 5). Full scripts and configuration files will be publicly available upon acceptance to ensure full reproducibility of our experimental results.
>
> We reiterate our gratitude for the reviewer’s support and their keen eye for presentation details. Addressing the "clear flaws" mentioned in the review has led to significant improvements in our work, which we hope can solidify your positive assessment. We remain fully open to any further discussion or suggestions.

---

> ### Comment · Reviewer_N9mv · 2025-11-24
>
> Thank you for your reply. Your changes significantly improve the paper. As far as I can tell Fig. 3 is significantly improved (unfortunately I no longer have access to the previous version). Furthermore, the additional experiments demonstrate clear utility over a previous benchmark.
>
> I remain with two final points:
> - (1) At least one citation (TopoBench) still reference the arXiv version. Please fix this.
> - (2) I would like to re-iterate that putting some visualizations of the generated graphs into the paper would clearly improve the paper. While I see that you are already short on space, maybe you can move less important things to the appendix.
>
> This is a good paper with a clear contribution and clear utility over previous. Thus, I will change my score to **accept**.

---

> > ### Author Response · Authors · 2025-11-26
> >
> > We sincerely thank the reviewer for the positive feedback and appreciation for the improvements we have made. This serves as strong motivation to perfect the paper as much as possible. Regarding the final two points:
> >
> > - We have corrected the arXiv citation reference - thank you for catching this and pointing it out. To the best of our knowledge, the remaining few arXiv papers have not been published yet to any proceedings.
> >
> > - We are currently making final adjustments to the paper as a result of ongoing discussions with other reviewers, including moving some minor sections between the main text and appendix. We hope to have made sufficient space by the end of the rebuttal period to include a small figure showing examples of generated graphs, which we completely agree would be a valuable addition to the paper.
> >
> > Once again, we are grateful for your help in strengthening the impact and presentation of our work. Please feel free to reach out with any further questions, comments or suggestions.

---

### Official Review · Reviewer_XXiP · 2025-10-31

**Soundness:** 3
**Presentation:** 4
**Contribution:** 3
**Rating:** 6
**Confidence:** 4

**Summary:**

This paper addresses the challenge of evaluating the true generalization capability of graph learning models by proposing the GraphUniverse framework. This framework systematically assesses model performance on unseen graph data by generating controllable graph families with semantically consistent communities. Experiments demonstrate that strong performance on known graphs does not guarantee generalization ability, and a model's robustness to distribution shifts heavily depends on the structural characteristics of the training graphs (such as the level of homophily in node connections), thus challenging the conventional practice of judging model quality solely based on transductive performance

**Strengths:**

1. The work directly confronts the "flawed benchmarking culture" in graph learning—an over-reliance on a few static benchmarks and transductive evaluation, which is considered a major bottleneck hindering progress in the field.

2. GraphUniverse is the first framework to enable large-scale, systematic evaluation of inductive generalization. Its paradigm of generating graph families with persistent semantics, rather than single graphs, is a significant advancement over existing tools like GraphWorld.

3. The experiments compellingly demonstrate the gap between transductive and inductive performance and reveal that model robustness is context-dependent. not universal. These findings have important implications for model evaluation and development in the field.

4. The paper is accompanied by an interactive demo and a promise to open-source the package. This is a valuable public contribution that will greatly facilitate future research.

**Weaknesses:**

1. The entire framework is built upon the Degree-Corrected Stochastic Block Model (DC-SBM), which is inherently biased toward generating graphs with strong community structures. This limits the applicability of the framework's conclusions to other common graph topologies that lack clear communities (e.g., geometric graphs, some biological networks), thus constraining its universality.

2. The experimental evaluation focuses primarily on node-level community detection and a simple graph-level task (triangle counting). It notably omits other critical graph learning tasks, especially Link Prediction, which is itself a core research area for inductive learning. This narrow task scope weakens the generality of the paper's main conclusions.

3. In light of the two points above, the paper's claim of enabling "systematic evaluation of inductive generalization" may be too broad. The findings and conclusions are largely about generalization for "node classification tasks" on "graphs with community structure," a scope limitation that should be more explicitly stated in the paper.

**Questions:**

1. How can the authors ensure that the paper's central finding—the gap between transductive and inductive performance—is not an artifact of the DC-SBM generator itself? Could models not designed for community-structured data be unfairly penalized by this evaluation framework?

2. Link prediction is a key inductive graph learning task. Why was it omitted from the benchmark? Can the GraphUniverse framework be extended or adapted to support a systematic evaluation of inductive link prediction?

---

> ### Author Response · Authors · 2025-11-20
> **Response to Reviewer XXiP**
>
> We sincerely thank the reviewer for their insightful comments and positive assessment of our paper's presentation and contribution. We are particularly grateful for the recognition of our framework as a significant advancement for systematic inductive evaluation and a direct answer to the "flawed benchmarking culture." We address below the valid concerns raised in their review:
>
> **On the Choice of DC-SBM and Its Implications.** The reviewer raises an important point about the implications of using the DC-SBM, whose formulation assumes discrete, non-overlapping communities. This was a conscious design decision that provided us with a direct, fine-grained, and interpretable control over the specific structural properties (homophily, degree heterogeneity, etc.) that are central to our systematic study of inductive generalization. However, this indeed doesn't capture all complex real-world phenomena like community overlap. Consequently, we included a detailed corresponding discussion in Appendix D to make this trade-off, and its potential limitations, explicit.
>
> Regarding the critical question of whether our findings are an artifact of the DC-SBM: our new real-world validation experiments (Subsection 5.5 / RQ4) help us partially address this. In this new analysis, we show that model rankings from GraphUniverse families achieve a consistent positive correlation with rankings on five diverse, real-world inductive datasets. This includes molecular benchmarks (e.g., OGBG-MolHIV, ZINC) whose underlying structure is not purely community-based, with the dataset tasks being provingly correlated to larger substructures of the graph (functional molecular groups) [1].
>
> This correlation suggests that the generalization gap we observe is not solely a by-product of the generator but reflects a genuine phenomenon captured by our framework. However, we agree this concern may still be valid for graph topologies that are poorly approximated by the DC-SBM, such as geometric graphs (further discussed in Appendix D). We are very open to any suggestions the reviewer might have on how to extend this validation or explore alternative generators.
>
> **On the Selection of Evaluation Tasks and the Omission of Link Prediction.** The reviewer makes a very fair point. We acknowledge that our current evaluation is focused on node and graph-level tasks, and the omission of link prediction is a notable limitation.
>
> This was a deliberate, practical trade-off. A primary goal of this paper was to demonstrate the broad utility of GraphUniverse across a comprehensive suite of modern graph models. To achieve this, we made the strategic decision to build our framework on top of the well-established TopoBench library. This integration allowed us to (1) easily plug in our generator, (2) benchmark against a large and diverse set of models, (3) ensure our experiments are highly reproducible, and (4) facilitate the usage of GraphUniverse as a benchmark for GNN and Topological Deep Learning practitioners.
>
> This decision came with one significant trade-off: TopoBench's current infrastructure is specialized for node and graph-level tasks and due to the codebase’s scale, costly to extend. While we plan to contribute a link prediction extension to TopoBench ourselves—as this is critical for our future work on Graph Foundation Models pre-training—we initially left it out of the scope of this initial paper. We are going to prioritize this extension the following days/weeks, though we recognize it may not be fully integrated before the end of the rebuttal period. Therefore, acting on the reviewer's feedback, for now we have carefully revised the abstract, introduction, and conclusion to more precisely state that our current claims are scoped to node and graph-level generalization–thus improving the rigor and clarity of our paper.
>
> We again thank the reviewer for their positive assessment and constructive feedback. We believe the concerns raised regarding the scope of our claims were spot-on, and addressing them has made our paper more precise and rigorous. We hope our responses and the corresponding revisions to the manuscript have clarified these points, and we would be very happy to engage in any further discussion.
>
> [1] Wu, Z., Wang, J., Du, H. et al. Chemistry-intuitive explanation of graph neural networks for molecular property prediction with substructure masking. Nat Commun 14, 2585 (2023). https://doi.org/10.1038/s41467-023-38192-3

---

> > ### Author Response · Authors · 2025-11-26
> >
> > We wanted to provide a brief update regarding our ongoing work during the rebuttal phase.
> >
> > Regarding **Link Prediction**: After a detailed assessment of the necessary engineering changes to the TopoBench infrastructure, we have confirmed that a robust integration will not be feasible before the rebuttal deadline. Consequently, we are maintaining our rigorous approach of explicitly scoping the paper’s claims to node and graph-level generalization in the revised text, ensuring the manuscript accurately reflects the current extensive evaluation.
> >
> > However, to further strengthen the framework's validation across diverse topologies (a concern you raised regarding the DC-SBM's bias), we are expanding our analysis of **low-homophily regimes**. As suggested by Reviewer *Wz29*, we are implementing and testing heterophily-specialized models (starting with FAGCN, we are sharing preliminary results in our response to that reviewer). We expect to upload a revised manuscript with the complete experiment shortly.
> >
> > We would like to thank the reviewer once again for their valuable feedback, and remain open to any further questions or suggestions.

---

> > > ### Author Response · Authors · 2025-11-28
> > >
> > > Following up on our discussion of heterophily-specialized models, we have now completed a full experiment (Appendix L) comparing FAGCN, H2GCN, and ChebNet across the homophily spectrum in both inductive and transductive settings. Notably, we find that H2GCN and ChebNet, which show expected superior performance in low-homophily transductive settings, lose this advantage entirely in the inductive setting. We hope this further validates our framework's ability to reveal paradigm-dependent architectural behaviors and highlights the importance of inductive evaluation for nuanced understanding of model capabilities.
> > >
> > > Once again, we appreciate the reviewer's thoughtful feedback.

---

### Official Review · Reviewer_Wz29 · 2025-10-31

**Soundness:** 2
**Presentation:** 3
**Contribution:** 2
**Rating:** 4
**Confidence:** 4

**Summary:**

This paper introduces GraphUniverse, a framework for generating families of synthetic graphs to systematically study inductive generalization in graph learning. By extending the Degree-Corrected Stochastic Block Model into a hierarchical generator, GraphUniverse produces semantically consistent graphs with controllable properties such as homophily, degree, and feature variance. Experiments show that model rankings differ substantially between transductive and inductive settings, revealing key insights into robustness and generalization behavior across diverse graph architectures.

**Strengths:**

* Fixes a fundamental limitation of GraphWorld and other synthetic graph generators by enabling multi-graph, inductive evaluation, moving beyond the conventional single-graph, transductive paradigm.
* Demonstrates that transductive performance is not predictive of inductive generalization and reveals architecture-specific robustness patterns under controlled distribution shifts. The analyses provide clear empirical evidence for a well-known but rarely quantified observation in the community—that graph models are often dataset- and domain-specific, performing well only in certain structural regimes. GraphUniverse’s controlled setup brings this behavior to light in a direct and systematic way.
* The inclusion of an interactive demo application is a valuable and user-friendly addition, enhancing reproducibility and making the framework easy to explore and adopt for future research.

**Weaknesses:**

* While the proposed framework convincingly demonstrates inductive generalization on synthetic graph families, it remains unclear whether these findings extend to real-world datasets. The paper does not map real datasets into the GraphUniverse parameter space or provide empirical validation that inductive robustness measured on synthetic graphs predicts robustness in real settings. As a result, it is difficult to determine whether the generated families capture realistic structural regimes or how the observed trends translate to practical graph learning scenarios.
* The claim that GraphUniverse can facilitate or benefit graph foundation model pretraining feels premature given the absence of supporting experiments or evidence. While this is not the paper’s central claim, the authors should at least include a more substantive discussion clarifying how GraphUniverse-generated data could be integrated into large-scale pretraining pipelines or what aspects of the framework make it suitable for such use.
* The underlying DC-SBM formulation assumes discrete, non-overlapping communities, whereas many real-world graphs contain overlapping structures and higher-order motifs that this model cannot capture.

My main concern is whether the insights obtained from GraphUniverse will transfer to real-world data. I suggest that the authors evaluate whether model rankings observed in the synthetic setting align with those on real benchmarks. For example, testing models on both heterophilic and homophilic real-world datasets could reveal whether performance trends from GraphUniverse are predictive of behavior in practical scenarios. It would be even more compelling if the authors also computed model rankings using GraphWorld and demonstrated that the rankings from GraphUniverse align more closely with those from real data. This would provide strong evidence for the external validity and practical relevance of the proposed framework. If the authors can demonstrate this, I am willing to increase my score.

**Questions:**

* Can the authors include a heterophily-specialized model (e.g., FAGCN, GPR-GNN, or H2GCN) in their comparisons, specifically for experiments varying homophily? This addition could better highlight how such models are specialized toward particular regimes by showing opposite performance trends across low- and high-homophily settings.
* Could the authors include a discussion of recent graph foundation model efforts that have demonstrated the benefits of using synthetic data, such as GraphFM \[1\], KumoRFM \[2\], and OpenGraph \[3\]? Including a brief overview of these works, along with parallels to how synthetic data has driven major advances in other domains, would strengthen the paper’s positioning and broaden its appeal to the wider graph learning and foundation model community.
* Can the authors add an analysis similar to Figures 1 and 2 from GraphWorld \[4\] to illustrate where real-world graph datasets lie within the GraphUniverse parameter space?

\[1\] Lachi, Divyansha, et al. "Graphfm: A scalable framework for multi-graph pretraining." *arXiv preprint arXiv:2407.11907* (2024).
\[2\] Fey, M., Kocijan, V., Lopez, F., Lenssen, J. E., & Leskovec, J. KumoRFM: A Foundation Model for In-Context Learning on Relational Data.
\[4\] Xia, L., Kao, B., & Huang, C. (2024). Opengraph: Towards open graph foundation models. *arXiv preprint arXiv:2403.01121*.

---

> ### Author Response · Authors · 2025-11-20
> **Response to Reviewer Wz29 (1/2)**
>
> We sincerely thank the reviewer for their thorough and constructive feedback. We appreciate the positive comments on the novelty of our framework for inductive evaluation and the utility of the interactive demo. We found the reviewer's main concerns and suggestions—particularly regarding the alignment with real-world datasets—to be actionable and highly valuable. We agree with the assessment and found these suggestions provided a clear path to strengthen the paper's contribution and soundness.
>
> **On the External Validity and Real-World Relevance of GraphUniverse.** We agree that demonstrating alignment with real-world datasets would significantly contribute to validating our framework. To directly address this concern, we have conducted a new set of experiments, which are now included in the paper as Subsection 5.5 (RQ4) and Appendix K. In this new analysis, we selected five real-world inductive graph-level datasets (OGBG-MolHIV, AOQSOL, NCI1, IMDB-MULTI, ZINC) and generated corresponding GraphUniverse families by matching the distributions of their key structural properties. We then trained our full suite of models and computed the correlation between the model rankings on the real datasets and their GraphUniverse equivalents (full details in Appendix K). Our results show a strong and consistent positive correlation, demonstrating that model performance and ranking trends observed in GraphUniverse are indeed predictive of inductive performance in real-world settings.
>
> **External Validity, Part 2: GraphWorld vs GraphUniverse.** Furthermore, we explicitly followed the reviewer’s suggestion to compare these results with GraphWorld. Therefore, for the same five datasets, we now also created single-graph GraphWorld equivalents (taking the mean of all graphs of all properties of the real dataset as target values, per the transductive paradigm) and compared their model rankings to the real-world inductive tasks and GraphUniverse equivalents. The rankings from the transductive GraphWorld setting show substantially lower, and in several cases negative, correlation with the real-world results. This new comparison provides strong empirical evidence for our paper's central claim: insights from the single-graph, transductive setting (GraphWorld) do not transfer to inductive generalization, whereas our multi-graph framework (GraphUniverse) provides a much stronger and more valid proxy. We hope this new analysis, highlighted as a new research question of Section 5 and fully detailed in Appendix K, addresses this main concern and provides evidence for our framework's external validity.
>
> **Clarifying the Positioning Regarding Graph Foundation Models.** We agree that the connection to Graph Foundation Models (GFMs) can be strengthened, and while this is not the paper's central claim, the potential is important to discuss. To address this, we have made two key additions to the revised manuscript.
> * First, as suggested, we have expanded our Related Work section to include a discussion of recent GFM efforts (such as GraphFM, KumoRFM, and OpenGraph) that demonstrate the value of diverse or synthetic data for pre-training. We use these examples to better contextualize our work, connecting it to the broader call within the community for better benchmarks and data generation tools to move beyond dataset-specific models—a core goal of the GFM paradigm.
> * Second, we have added a new, more substantive discussion in Appendix B that elaborates on the two primary roles we envision for GraphUniverse:
>     * As a prototyping and evaluation testbed: GraphUniverse allows for the rapid generation of limitless, diverse graph families for pre-training, followed by systematic evaluation on fine-tuning tasks with known and controlled distribution shifts (e.g., varying homophily, density, or degree). This enables a precise, low-cost method for benchmarking a GFM's robustness, directly addressing the concerns raised in works like _"Position: Graph learning will lose relevance due to poor benchmarks"_ regarding the limitations of current evaluation practices.
>     * As an extensible data augmentation engine: We also discuss the framework's potential to be extended (beyond the scope of this initial paper) to fit fine-grained, community-specific properties of real datasets. This would allow it to generate novel but realistic 'in-domain' graph instances, serving as a powerful data augmentation tool to improve GFM robustness—a strategy highlighted as valuable in recent surveys like _"Graph foundation models: A comprehensive survey"_.
>
> (1/2)

---

> > ### Author Response · Authors · 2025-11-20
> > **Response to Reviewer Wz29 (2/2)**
> >
> > **Acknowledging the Limitations of the DC-SBM Generator.** The reviewer rightly notes the inherent limitations of the DC-SBM, providing the opportunity for an interesting discussion. As we see it, these limitations can be broadly categorized into two types:
> > * _Easily extensible limitations (design choices for simplicity):_ Features like deterministic community membership and non-overlapping communities could readily be implemented as extensions. We chose these simplifications to maintain experimental interpretability and focus on core phenomena that are clearly identifiable, controllable and translatable to real-world settings. For researchers studying specific aspects like overlapping communities, our framework provides a foundation for such extensions.
> > * _Fundamental limitations (higher-order structures):_ We acknowledge that our framework cannot directly control complex motif-driven structures (e.g., specific triangular patterns, star motifs, geometric arrangements). This represents a significant limitation and an interesting avenue for future work. Nevertheless, as we already discussed before, our RQ4 experiments demonstrate that despite these limitations, GraphUniverse-generated datasets effectively predict real-world model performance, achieving strong correlation with real datasets. This is especially relevant, knowing that some of the tasks in our real-world validation datasets are dependent on higher-order structures, such as chemical property prediction being dependent on complex functional groups and global structural features [1]. We hope that the transferability we observe suggests that community-centric evaluation captures sufficient fundamental graph learning capabilities—the interplay between local structure, features, and connectivity patterns—for insightful model assessment.
> >
> > We have now added a detailed discussion in the revised manuscript in Appendix D to make this trade-off explicit, noting that integrating more complex generators is a valuable direction for future extensions.
> >
> > **On Heterophily-Specific Models and Real-World Dataset Visualization.** We thank the reviewer for these two suggestions for extending our analysis. Indeed, including a heterophily-specialized model (e.g., FAGCN) would provide a valuable contrast in our homophily experiments. While we prioritized the new real-world validation experiments (RQ4) for this response, we have now implemented the FAGCN model in the TopoBench benchmark and are starting to run the homophily experiments using it. We plan to include these results in the revised manuscript before the rebuttal deadline–we will keep you updated when we do. Regarding the suggestion to create plots similar to GraphWorld's (Figures 1 & 2), this is a fascinating but non-trivial request. Visualizing a single graph's mean properties (as in GraphWorld) is straightforward, but meaningfully plotting a distribution of real-world graphs within the multi-dimensional parameter space of GraphUniverse families is conceptually challenging. We are actively considering how to best represent this, but we find it difficult to condense into an insightful 2 or 3D plot. We would be genuinely grateful for any further ideas the reviewer might have on a suitable visualization for this inductive setting.
> >
> > Once again, we appreciate the reviewer’s detailed and insightful guidance, which has helped us strengthen Graphuniverse contribution and validity. We hope our revisions addressed their initial concerns, and would be happy to discuss any of these points further.
> >
> > [1] Wu, Z., Wang, J., Du, H. et al. Chemistry-intuitive explanation of graph neural networks for molecular property prediction with substructure masking. Nat Commun 14, 2585 (2023). https://doi.org/10.1038/s41467-023-38192-3
> >
> > (2/2)

---

> > > ### Comment · Reviewer_Wz29 · 2025-11-26
> > >
> > > I thank the authors for the detailed and thoughtful rebuttal and for the substantial revisions made to the manuscript. The new experiments on real-world validation and the comparison with GraphWorld directly address my primary concern and significantly strengthen the paper’s central claims.
> > >
> > > First, it would be helpful if the appendix included a short explanation of how deterministic community membership and non-overlapping communities could be extended in future work. I am not requesting additional experiments, but a brief description of how these extensions could be implemented would make the framework more accessible for researchers who wish to build on it.
> > >
> > > Second, the comparison between GraphWorld and real-world evaluation is an important piece of evidence for the paper’s claims. I recommend moving part of this comparison into the main text, since it strengthens the overall argument.
> > >
> > > I also look forward to seeing the FAGCN results.
> > >
> > > Overall since my main concern has now been addressed, I am raising my score to 6 and support acceptance of this paper in its current state. I am looking forward to the authors responses to my minor concerns above.

---

> > > > ### Author Response · Authors · 2025-11-26
> > > >
> > > > We sincerely thank the reviewer for their encouraging response, we are very happy that the real-world validation experiments have addressed the primary concerns.
> > > >
> > > > While we address your latest suggestions—we will get back to you shortly with a full response and the corresponding manuscript revisions—we wanted to immediately share the initial results for FAGCN across varying homophily levels, as promised:
> > > >
> > > > https://postimg.cc/2LRK6G32
> > > >
> > > > As shown, FAGCN performance does not stand out in heterophilic regimes; it outperforms GCN in the inductive setting, but surprisingly it does not perform well in the transductive one. This nuance highlights the complexity of heterophilic generalization and aligns with the need for the granular evaluation our framework provides.
> > > >
> > > > Given the observed results, we are currently extending the experiments for other heterophilic models and will upload a revised manuscript incorporating these full results—along with the improvements you just requested—shortly.

---

> > > > > ### Author Response · Authors · 2025-11-28
> > > > >
> > > > > We are pleased to provide the complete follow-up response with the full experimental results and manuscript revisions we indicated.
> > > > >
> > > > > **On the Extensibility Documentation.** We have extended our original discussion in Appendix D to now include a concrete proposal on how one could implement community overlap within our framework (Appendix Section D.1.1). We agree that having such a concrete example in our appendix will guide researchers using our framework, both for this particular extension and as inspiration for other types of extensions.
> > > > >
> > > > > **Regarding Heterophily-specialized Models.** Regarding heterophily-specialized models: We have added a new full experiment in Appendix L which compares three heterophily-specialized models (FAGCN, H2GCN, and ChebNet) to our existing suite across the homophily spectrum in both inductive and transductive settings. **Most notably, we find that H2GCN and ChebNet, which demonstrate clear superior performance in low-homophily regimes in the transductive setting, lose this advantage in the inductive setting.** FAGCN notably underperforms in both settings, suggesting its frequency-adaptive mechanisms may not translate effectively to our synthetic community detection task. This paradigm-dependent behavior highlights the importance of evaluating models in inductive settings, particularly given that many specialized architectures are primarily validated in transductive scenarios that may be more prone to overfitting to specific graph structures. Our results demonstrate the value of systematic evaluation across different graph properties and settings to gain a comprehensive understanding of when and why architectural advantages emerge.
> > > > >
> > > > > **On Moving the GraphWorld Comparison to the Main Text.** We agree with the reviewer that this comparison is central to our argument, but are unsure what specific aspect they are referring to—the text of RQ4 in the main body (together with Figure 5) already summarizes the main details and key findings of the GraphWorld vs. GraphUniverse comparison. We would have kindly asked for further guidance about the particular details of Appendix K that the reviewer wanted to find in the main text.
> > > > >
> > > > > We thank the reviewer again for this constructive dialogue and would have loved to remain available to continue this interesting discussion.

---

### Author Response · Authors · 2025-11-20
**General Response to All Reviewers**

We are genuinely grateful for the quality and depth of feedback provided by all reviewers. Their constructive criticism and insightful suggestions have improved our paper's clarity, rigor, and impact. Based on the common concerns raised, we have made revisions to the manuscript (highlighted in blue for easy identification) that we hope strengthen our contribution.

**Main Updates**

* **Real-World Validation (RQ4):**  In response to multiple concerns about synthetic-to-real transferability, we added comprehensive experiments demonstrating that GraphUniverse effectively predicts real-world model performance across five diverse datasets, achieving strong positive correlations while GraphWorld equivalents often show negative correlations.
* **Code Release:** All GraphUniverse code is now available in our anonymous repository: https://anonymous.4open.science/r/GraphUniverse-3458
* **Detailed Discussion on DC-SBM Biases and Foundation Model Claims:** We expanded discussion of both our generator's limitations and foundation model claims in the main text, with detailed analysis in Appendices B and D.

We welcome continued discussion and remain committed to further revisions that strengthen our contribution to the field. Detailed responses to individual reviewer comments follow below.

---

### Meta-Review · Area_Chair_hD2E · 2026-01-07

**Summary:**

This paper proposes GraphUniverse, a hierarchical generator of graph families that can enable systematic evaluation of inductive generalization in graph learning.  The authors aim to go beyond single graph, transductive benchmarks which the community largely uses.  Reviewers found the contribution timely and impactful, and liked the conceptual shift to multi-graph inductive evaluation, the experiments revealing sharp trans and inductive gaps, and the real-world experiments showing that GraphUniverse rankings correlate well with real inductive benchmarks and outperform GraphWorld as a proxy (Wz29, XXiP, N9mv, 3ZL3). The interactive demo, code release, and improved presentation further strengthened the work (Wz29, N9mv). Main reviewer concerns focused on reliance on DCSBM generator and inheriting of its biases for community-structured graphs, the initial breadth of claims (especially re GFMs and prominence), the omission of link prediction as a benchmarking area, and some presentation issues (XXiP, N9mv, 3ZL3).  However, reviewers addressed several of these through better scoping of claims, expanded discussion of limitations, and additional experiments which led multiple reviewers to update their score.  Overall, this work leans towards acceptance.

**Reviewer Concerns:**

See above.

**Reviewer Scores:**

Wz29: 4 -> 6
XXiP: 6->6
N9mv: 6->6/7
3ZL3: 4->4/5

---

### Decision · Program_Chairs · 2026-01-26

Accept (Poster)